# Characterization and structural basis of a lethal mouse-adapted SARS-CoV-2

Shihui Sun [1,7], Hongjing Gu[1,7], Lei Cao[2,7], Qi Chen[1,7], Qing Ye[1,7], Guan Yang[3,7], Rui-Ting Li[1,7], Hang Fan [1,7], Yong-Qiang Deng [1], Xiaopeng Song[3], Yini Qi[3], Min Li[1], Jun Lan[2], Rui Feng[2], Yan Guo[1], Na Zhu [4], Si Qin[1], Lei Wang[2], Yi-Fei Zhang[1], Chao Zhou [1], Lingna Zhao[1], Yuehong Chen[1], Meng Shen[1], Yujun Cui[1], Xiao Yang [3], Xinquan Wang [5], Wenjie Tan [4], Hui Wang[1✉], Xiangxi Wang [2✉] & Cheng-Feng Qin [1,6✉]

There is an urgent need for animal models to study SARS-CoV-2 pathogenicity. Here, we generate and characterize a novel mouse-adapted SARS-CoV-2 strain, MASCp36, that causes severe respiratory symptoms, and mortality. Our model exhibits age- and gender-related mortality akin to severe COVID-19. Deep sequencing identified three amino acid substitutions, N501Y, Q493H, and K417N, at the receptor binding domain (RBD) of MASCp36, during in vivo passaging. All three RBD mutations significantly enhance binding affinity to its endogenous receptor, ACE2. Cryo-electron microscopy analysis of human ACE2 (hACE2), or mouse ACE2 (mACE2), in complex with the RBD of MASCp36, at 3.1 to 3.7 Å resolution, reveals the molecular basis for the receptor-binding switch. N501Y and Q493H enhance the binding affinity to hACE2, whereas triple mutations at N501Y/Q493H/K417N decrease affinity and reduce infectivity of MASCp36. Our study provides a platform for studying SARS-CoV-2 pathogenesis, and unveils the molecular mechanism for its rapid adaptation and evolution.

[1] State Key Laboratory of Pathogen and Biosecurity, Beijing Institute of Microbiology and Epidemiology, AMMS, Beijing 100071, China. [2] CAS Key Laboratory of Infection and Immunity, National Laboratory of Macromolecules, Institute of Biophysics, Chinese Academy of Sciences, Beijing 100101, China. [3] State Key Laboratory of Proteomics, Beijing Proteome Research Center, National Center for Protein Sciences (Beijing), Beijing Institute of Lifeomics, Beijing 102206, China. [4] National Institute for Viral Disease Control and Prevention, Chinese Center for Disease Control and Prevention (China CDC), Beijing 102206, China. [5] The Ministry of Education Key Laboratory of Protein Science, Beijing Advanced Innovation Center for Structural Biology, Beijing Frontier Research Center for Biological Structure, Collaborative Innovation Center for Biotherapy, School of Life Sciences, Tsinghua University, Beijing 100084, China. [6] Research Unit of Discovery and Tracing of Natural Focus Diseases, Chinese Academy of Medical Sciences, Beijing 100071, China. [7] These authors contributed equally: Shihui Sun, Hongjing Gu, Lei Cao, Qi Chen, Qing Ye, Guan Yang, Rui-Ting Li, Hang Fan. ✉email: geno0109@vip.sina.com; xiangxi@ibp.ac.cn; qincf@bmi.ac.cn

Coronavirus disease 2019 (COVID-19) caused by severe acute respiratory syndrome coronavirus 2 (SARS-CoV-2) has resulted in a public health crisis[1]. The symptoms of COVID-19 are similar to those of SARS-CoV and MERS-CoV infections, ranging from fever, fatigue, dry cough and dyspnea, and mild pneumonia to acute lung injury (ALI) and the acute respiratory distress syndrome in severe cases. In fatal cases, multi-organ failures accompanied by a dysregulated immune response have been observed[2–4]. Numerous studies have highlighted age- and gender-related discrepancies in the distribution of COVID-19 cases where the elderly and men tend to have a higher case–fatality ratio when compared to the young and females, suggesting that elderly man are more likely to succumb to COVID-19[5,6].

Similar to SARS-CoV, SARS-CoV-2 belongs to the *Betacoronavirus* genus of the *Coronaviridae* family, and is an enveloped, single-stranded positive-sense RNA virus. Human angiotensin-converting enzyme 2 (hACE2) has been demonstrated as the functional receptor for SARS-CoV-2[7,8]. SARS-CoV-2 cannot infect standard laboratory mice due to inefficient interactions between the receptor-binding domain (RBD) of Spike (S) protein and mouse ACE2 (mACE2)[9]. So, several hACE2-expressing mouse models such as hACE2 transgenic mice[10], AAV-hACE2 transduced mice[11], and Ad5-hACE2 transduced mice[12] have been developed. Furthermore, mouse-adapted strains of SARS-CoV-2 have also been developed via either in vivo passaging or reverse genetics[13–15]. However, most these models cause only mild to moderate lung damage in mice. A small animal model capable of recapitulating the most severe respiratory symptoms and high case–fatality ratio of COVID-19 remains of high priority.

In this study, we generated a mouse-adapted strain, MASCp36, from a previous described, MASCp6, by further in vivo passaging for additional 30 times in mice[13]. Further characterization demonstrated MASCp36 caused 100% fatality in 9-month-old, male BALB/c mice with severe malfunctions of the respiratory system and multi-organ damage. Combined biochemical assay, viral genome sequencing, and cryo-EM analysis clearly demonstrated the critical role of the progressively emerged amino acid mutations in the RBD of the mouse-adapted strains at different passages.

## Results

**Generation of a lethal mouse-adapted strain of SARS-CoV-2.** In our previous study, we generated a mouse-adapted strain of SARS-CoV-2 (MASCp6) by six serial passages of a SARS-CoV-2 clinical isolate in the lung of BALB/c mice, which caused moderate lung damage in mice. Herein, we further serially passaged for additional 30 times to generate a more virulent mouse-adapted strain, and the resulting SARS-CoV-2 at passage 36 (named as MASCp36) was used for stock preparation and titration.

To characterize the pathogenicity of MASCp36 to standard laboratory mice, groups of BALB/c mice at different age and sex were subjected to intranasal injection of varying doses of MASCp36. Strikingly, survival curve analysis showed that 9-month-old mice are highly susceptible to MASCp36 challenge, and the infected animals succumbed to MASCp36 challenge in a dose-dependent manner. Moreover, male mice were more susceptible to MASCp36 in comparison to female ones, and the 50% lethal dose (LD50) was calculated to 58 plaque forming unit (PFU) (male) and 690 PFU (female), respectively (Fig. 1a, b). All 9-month-old mice challenged with high doses (1200 or 12,000 PFU) of MASCp36 developed typical respiratory symptoms and exhibited features like ruffled fur, hunched back, and reduced activity. Of particular note, tachypnea was common in all moribund animals. In addition, this unique gender-dependent

mortality was also recorded in 9-month-old C57BL/6 mice challenged with MASCp36 (Supplementary Fig. 1). Interestingly, 8-week-old mice, either male or female, were resistant to MASCp36 challenge, and only one animal that received 12,000 PFU of MASCp36 challenge died during the observation period (Fig. 1c, d). Thus, the developed MASCp36 was sufficient to cause mortality in BALB/c or C57BL/6 mice in an age- and gender-skewed manner.

We further characterized the in vivo replication dynamics and tissue distribution of MASCp36 in mice. The results from qRT-PCR showed that high levels of SARS-CoV-2 subgenomic RNAs were persistent in the lung and tracheas till 4 days post infection (dpi) in 9-month-old mice (Fig. 1e). Interestingly, the 8-week-old mice sustained a similar tissue distribution as the 9-month-old ones upon MASCp36 challenge, and lung and tracheas represented the major tissues supporting viral replication (Fig. 1f). Consistent with the sgRNA levels, infectious SARS-CoV-2 could be recovered in the lung homogenate from 9-month-old and 8-week-old mice that received MASCp36 challenge (Fig. 1g, h). Multiplex immunofluorescence staining showed that large amount of SARS-CoV-2 N protein positive signals were detected in lung sections from the MASCp36-infected mice (Fig. 2a, b), and co-localization with the SPC+ alveolar type 2 (AT2) cells and CC10+ club cells, as well as the ACE2+ club cells, was shown (Fig. 2a), while few FOXJ1+ ciliated cells and PDPN+ alveolar type 1 (AT1) cells were infected with SARS-CoV-2 (Fig. 2a). More importantly, despite more SPC+AT2 cells were detected in the lung of younger mice, SARS-CoV-2-infected cells were more abundant in the lung from 9-month-old mice than those from the 8-week-old animals (Fig. 2b). In fact, the ratio of ACE2+ AT2 cells in the uninfected 9-month-old mice was much higher than that in the uninfected 8-week-old mice (Fig. 2c), which supported the observed age-skewed susceptivity to SARS-CoV-2.

**MASCp36 causes multiple organs damage in mice.** To further characterize the pathological outcome in MASCp36-infected mice, lung tissues were collected at 4 dpi and subjected to histopathological and immunostaining analysis. Naked eye observation recorded severe lung injury characterized with bilateral cardinal red appearance and sticky mucus in lungs when compared with that from the uninfected control animals (Fig. 3a). According to the metrics of ALI laid out by the American Thoracic Society[16], MASCp36 infection induced necrotizing pneumonia and extensive diffuse alveolar damages (DAD) on 4 dpi. The microscopic observation showed large quantities of desquamative epithelial cells in bronchiole tubes (yellow arrow) and a large area of necrotic alveoli epithelial cells, fused alveoli walls with inflammatory cells infiltration especially neutrophils in alveolar septae or alveolar airspace, serious edema around vessels (cyan arrow) and scattered hemorrhage (blue arrow) (Fig. 3a). In addition, foamy cells, polykaryocytes, fibrin cluster deposition, and hyaline membrane formation were common in the MASCp36-infected animals (Fig. 3b), indicative of ALI, which is well characterized in severe COVID-19 patients[4]. Besides, the formation of typical viral inclusion bodies was also observed (Fig. 3b). In addition, we also observed the lung pathology in 9-month-old male mice infected with a lower dose (120 PFU) of MASCp36 at 7 dpi. As shown in Fig. 3c, lung fibrosis was observed in the MASCp36-infected mice as evidenced by the depositions of collagen in pulmonary artery wall and thickened alveoli. Meanwhile, an increased proliferation of alveolar mesenchymal cells was also observed in MASCp36-infected mice (Supplementary Fig. 2).

Apart from the lung damage, spleen damage was also observed in MASCp36-infected mice, which was characterized by atrophic

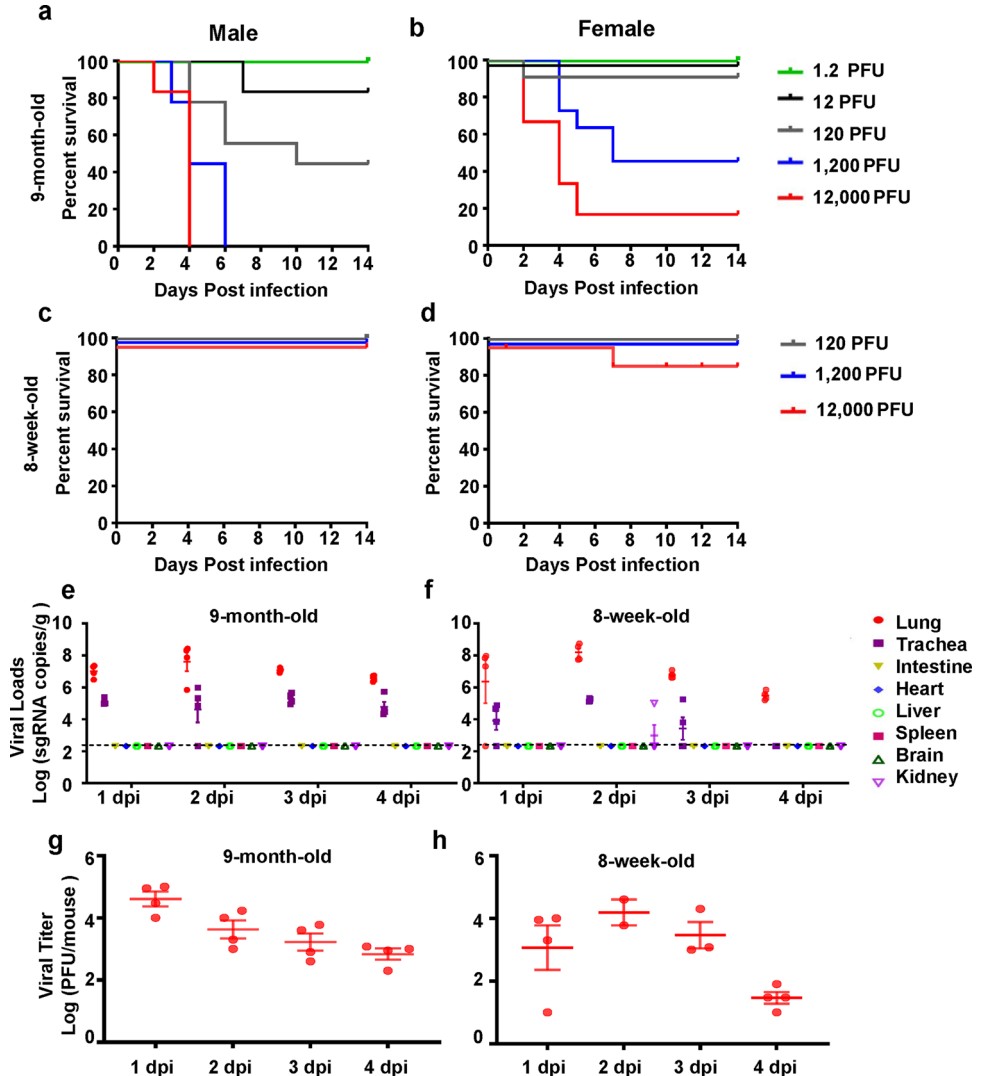

**Fig. 1 MASCp36 is highly virulent in 9-month-old mice. a–d** Survival curve of BALB/c mice upon challenge with MASCp36. Groups of female or male 9-month-old BALB/c mice and 8-week-old BALB/c mice were infected intranasally with the indicated doses of MASCp36, and the clinical symptoms and mortality were recorded for 14 days (n ≥ 6 per group). **e, f** Replication dynamics and tissue distribution of SARS-CoV-2 sgRNAs in mice infected with MASCp36. Groups of 9-month-old BALB/c mice were i.n. inoculated with 12,000 PFU of MASCp36, and sacrificed at 4 dpi. All the indicated tissue samples were collected and subjected to viral sgRNA load analysis by qRT-PCR. Dash lines denote the detection limit. Data are presented as means ± SEM (n = 4 per group). **g, h** Infectious viral titers in lung tissues in mice infected with MASCp36 were detected by plaque formation assay. Data are presented as means ± SEM (n = 4 per group).

splenic corpuscle, splenic cells necrosis and hemorrhage in red pulp (green arrow) (Fig. 4a). Multiplex immunofluorescence staining showed a striking loss of germinal centers in the spleen as suggested by the reduction in CD19+ B and CD3+ T cell counts as well as diminished ICOS+ follicular helper T cells (Fig. 4b). Similar observations have been well described in severe COVID-19 patients[17]. In addition, renal tubular damage with casts in renal tubules, as well as breakdown of CD31+ glomerular capillary epithelium and PDPN+ basement membrane, was also observed in kidney (Fig. 4c, d). Similar splenic lesion and kidney damage were also reported in cases with postmortem examinations of deceased COVID-19 patients[18,19].

**MASCp36 induces age- and gender-specific response in mice.** To characterize the host response to MASCp36 infection in BALB/c mice at different age and gender, RNA-Seq analysis was performed using lung homogenates collected at 1 and 4 dpi. Upon MASCp36 infection, large numbers of genes were readily regulated in both 8-

week-old and 9-month-old mice at 1 dpi (Fig. 5a). In the MASCp36-infected 9-month-old mice, 16 of the top 20 upregulated genes were interferon (IFN)-simulated genes (ISGs) and cytokines, including Cxcl10, Mx1, Gbp7, Mx2, Ifi44, Oasl2, Itgp, Gbp9, Isg15, Ifit1, Gbp4, Irgm2, Irf7, Ifit2, and Stat2, while in the 8-week-old mice, only Ifit3, Mx2, Trim34b, Ifi44l, and Rtp4 were related to antiviral immunity. At 4 dpi, the 9-month-old mice exhibited more dramatic response with a total of 2762 upregulated genes and 2363 downregulated genes. And cytokines Il6 and Ccl2 were among the top 20 upregulated genes at 4 dpi, while ISGs including Mx1, Ifit1, Ifit3, Irf7, Dhx58, and Ifi44 were in the MASCp36-infected 8-week-old mice (Fig. 5a). Gene Ontology (GO) enrichment analyses showed that MASCp36 infection induced strong transcription of genes related to antiviral response, inflammatory, cytokine production, and cell death in both 8-week-old and 9-month-old mice infected with MASCp36 (Fig. 5b). Strikingly, the induction of "cytokine production," "leukocyte activation," "inflammatory response," "interferon-gama production," and "positive regulation of cell death" was much quicker and stronger in the older mice. Of particular note, a large

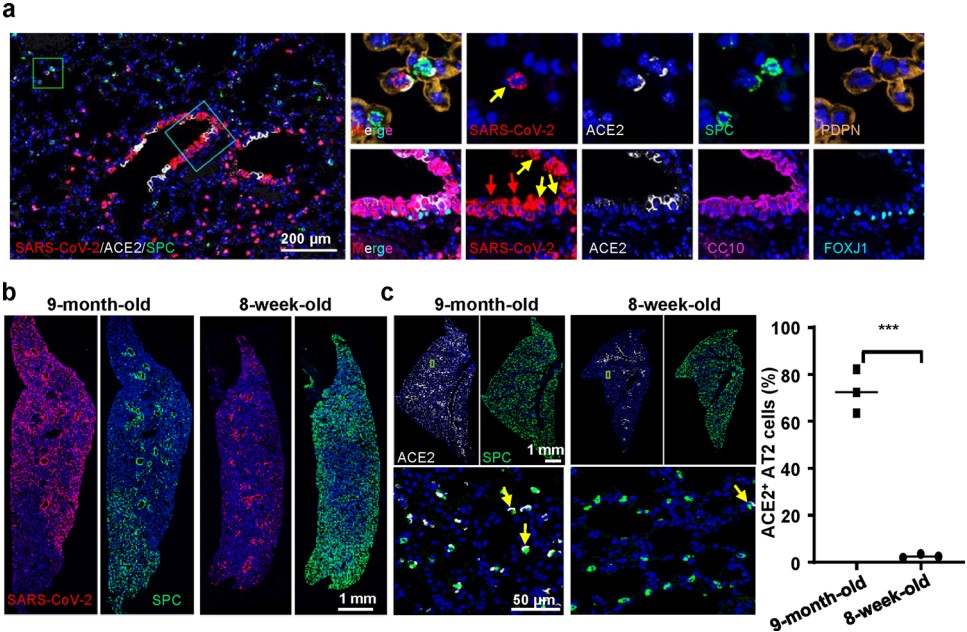

**Fig. 2 Difference of MASCp36 cellular tropism in lungs from 9-month-old and 8-week-old mice. a** Multiplex immunofluorescence staining for detection of SARS-CoV-2 targeted cells in lung sections of male mouse (9-month-old) at 1 dpi. SARS-CoV-2 N protein (red), ACE2 (white), SPC (green), PDPN (gold), CC10 (magenta), FOXJ1 (cyan). The framed areas are shown adjacently at a higher magnification. The yellow arrows indicate ACE2+ club cells infected with SARS-CoV-2, and the red arrows indicate decreased ACE2 expression in infected cells ($n = 3$ per group). **b** Multiplex immunofluorescence staining of lung sections for SARS-CoV-2 N protein (red) and SPC (green) detection in male mice at 1 dpi ($n = 3$ per group). **c** Multiplex immunofluorescence staining for detection of ACE2 (white) and SPC (green) expression in lung tissues from the uninfected control mice (9-month-old and 8-week-old). The framed areas are shown below at a higher magnification. The yellow arrows indicate ACE2+ cells. The percentage of ACE2+ cells in the SPC+ AT2 compartment was statistically analyzed by two-tailed Student's t test ($P = 0.0002$) ***$P < 0.01$. Data are presented as mean ± SD ($n = 3$ per group).

number of cytokines and chemokines transcription were upregulated in response to MASCp36 in 9-month-old mice, while less and smaller upregulation of cytokines and chemokines were observed in 8-week-old mice. Ccl2, Ccl7, Il6, Cxcl10, and Cxcl11 were the most significant upregulated genes in the MASCp36-infected 9-month-old mice (Fig. 5c). Meanwhile, Luminex analysis also detected obvious elevated cytokine and chemokine production, including IL6, CCL7, CCL12, CXCL10, CXCL16, CCL3, CXCL1, and CXCL13, in the lung homogenates from the MASCp36-infected mice (Fig. 5d and Supplementary Fig. 3). Strikingly, we found that many genes involved in "cilium movement" were significantly stimulated at 1 and 4 dpi in the younger mice, while only a small number of genes were upregulated in the older mice at 4 dpi (Fig. 5e).

In addition, immunofluorescence staining of lung sections also indicated that MASCp36 infection caused more extensive cell death and AT2 loss in the 9-month-old mice (Fig. 6a). Interestingly, less CD68+ macrophages and Ly-6G+ neutrophils were detected in 9-month-old mice than that in young mice 1 dpi (Fig. 6b), and reversed on 4 dpi. Of note, although the inflammatory cell infiltration was lagged on day 1 in 9-month-old mice, it elevated and sustained a high level until 4 days. However, the response was rapid and short-lived in young mice. The results indicated that the lagged and sustained immune response to viral infection may be related to the lung damage and contribute to the more severe outcomes in 9-month-old mice.

Furthermore, to understand the gender-skewed mortality in mice, we also compared the different gene expression in lungs of male and female mice. First of all, we confirmed that 1075 genes were significantly highly expressed in female mice (Supplementary Fig. 4a) in total of 1308 differentially expressed genes (DEGs), and the genes enriched in "Cilium movement," "Immunoglobulin production," "Adaptive immune responses," and "Cellular response to interferon-beta" GO terms, which

coincided with previous studies[20]. After infection, both the antiviral response and inflammatory response were stronger elevated in female mice when compared to the male especially on 1 dpi (Supplementary Fig. 4b, c).

**H014 confers full protection in lethal MASCp36 model.** To further test the utility of this MASCp36-based mouse model for evaluation of countermeasures, H014, a known human monoclonal antibody (mAb) targeting the RBD of SARS-CoV-2[21], was examined for its ability to prevent mortality caused by MASCp36. Administration of a single dose of H014 (50 mg/kg) resulted in 100% survival (Fig. 7a) and complete clearance of virus from the lungs of infected mice (Fig. 7b, c). By contrast, all animals that received a control isotype mAb treatment showed high levels of viral replication in lungs and eventually died with respiratory diseases within 5 days (Fig. 7a–c). Multiplex immunofluorescence staining demonstrated H014 treatment completely protected animals from viral infection and no viral antigen was detected at 4 dpi, while significant AT2 loss and neutrophil infiltration were seen in the mice treated with isotype mAb (Fig. 7c). Hematoxylin and eosin (H&E) staining also confirmed that H014 treatment dramatically prevented the MASCp36-induced lung damage characterized by fused alveoli walls, desquamative epithelial cells, severe edema, and scattered hemorrhage (Fig. 7d). These data perfectly match the results from hACE2 humanized mice[21], highlighting the potential of MASCp36-based models in evaluating the in vivo therapeutic efficacy.

**Adaptive mutations responsible for the enhanced binding affinity to ACE2.** To deduce the genetic basis for the lethal phenotype of MASCp36, deep sequencing was performed to identify the mutations emerged during the in vivo passaging history. Sequence

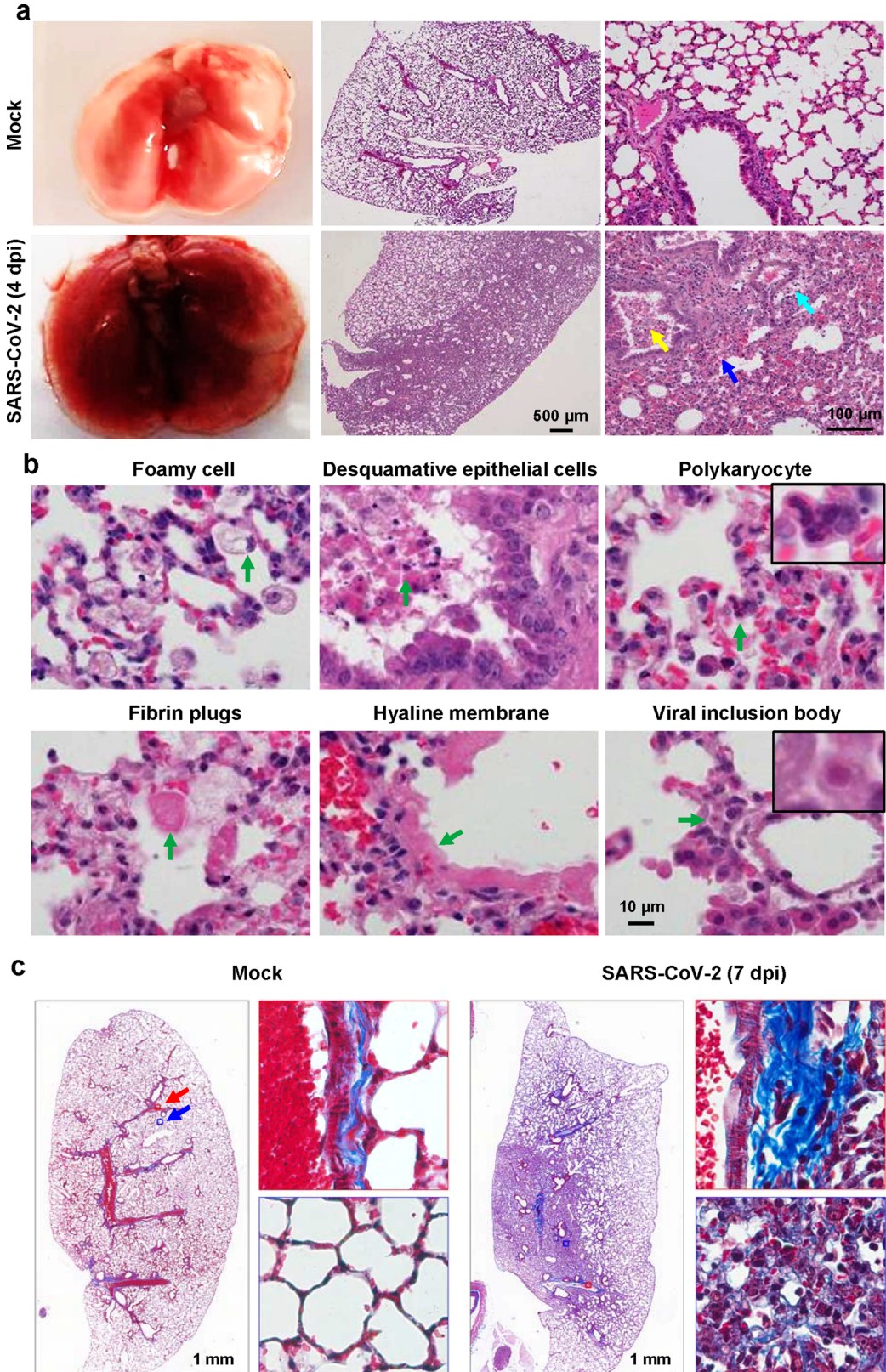

**Fig. 3 Acute lung damage caused by MASCp36 infection in mice. a** Gross necropsy and hematoxylin and eosin (H&E) staining of lung sections from male BALB/c mice (9-month-old) infected with 1200 PFU of MASCp36. Yellow arrow indicates desquamative epithelial cells in bronchiole tubes, cyan arrow indicates edema around vessels, and blue arrow indicates hemorrhage. Representative images are shown ($n = 3$ per group). **b** Microscopic observation of lungs showing foamy cells, desquamative epithelial cells, polykaryocytes, fibrin plugs, hyaline membrane, and viral inclusion body ($n = 3$ per group). **c** Masson's trichrome staining of lung sections from male BALB/c mice (9-month-old) infected with 120 PFU of MASCp36 ($n = 5$ per group) at 7 dpi. Framed areas are shown adjacently at higher magnification.

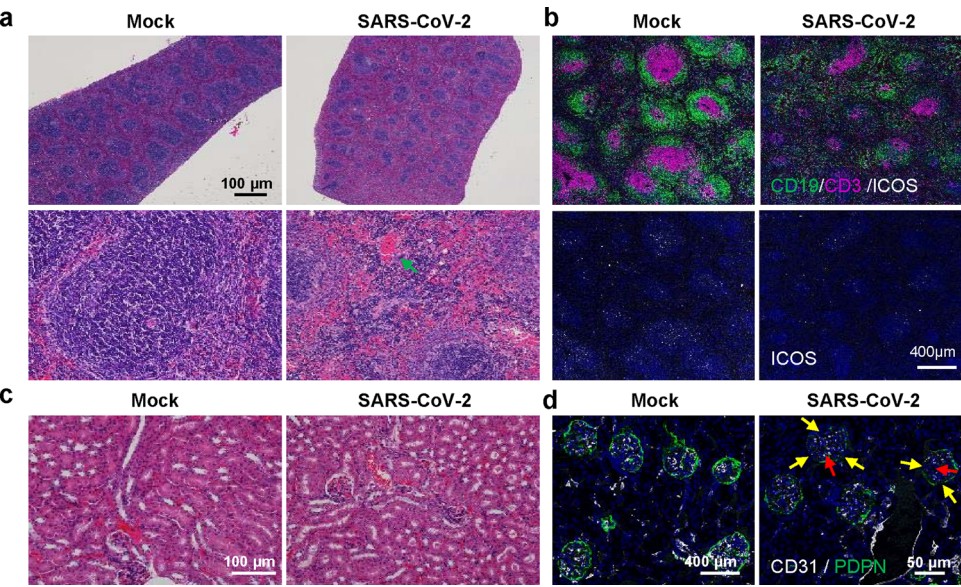

**Fig. 4 Spleen and kidney injury during BALB/c mice infection with MASCp36. a** H&E staining of spleen sections from male BALB/c mice (9-month-old) infected with MASCp36 (n = 3 per group). **b** Multiplex immunofluorescence staining of mouse spleen sections for detection of CD19 (green) B cells, CD3 (magenta) T cells, and ICOS (white) follicular helper T cells (n = 3 per group). **c** H&E staining of kidney sections from male BALB/c mice (9-month-old) infected with MASCp36 at 4 dpi (n = 3 per group). **d** Multiplex immunofluorescence staining of mouse kidney sections for CD31(white) and PDPN (green) (n = 3 per group). The yellow and red arrows indicate discrete glomerular basement membrane and endothelial vessels, respectively.

comparisons of the WT strain and mouse-adapted strains at different passages (6, 15, 25, 30, and 36) revealed a process of gradual accumulation of amino acid substitutions (Fig. 8a). Besides the four mutations (L37F, P84S, N501Y, and D128Y) identified in MASCp6, MASCp36 acquires additional eight amino acid substitutions, including I1258V, H470Y, S301L, A128V, S8F, K417N, Q493H, and R32C in the NSP3, NSP4, NSP5, NSP6, NSP7, S, and N (Fig. 8a). Specially, single (N501Y), double (Q493H, N501Y), and triple (K417N, Q493H, N501Y) mutations in the RBD were identified in MASCp6, MASCp25, and MASCp36, respectively (Fig. 8b). To clarify the potential role of these mutations, the RBD of these different adaptive strains were expressed in HEK Expi 293F cells, and their binding affinities to mACE2 were determined by surface plasmon resonance (SPR). As expected, the RBD of WT SARS-CoV-2 presented no detectable binding, but RBDs from different passages of mouse-adapted strains (RBD$_{MASCp6}$, RBD$_{MASCp25}$, and RBD$_{MASCp36}$) gain gradually enhanced binding abilities to mACE2 with affinities ranging from ~500 to 2 μM (Fig. 8c). Meanwhile, we also compared the binding characterization of these RBD mutants and hACE2. As shown in Fig. 8d, all three RBD mutants could bind to hACE2 with affinities at nanomole (ranging from 500 to 20 nM). Interestingly, compared to RBD$_{MACSp6}$ with a mutation of N501Y, RBD$_{MACSp25}$, which contains an extra Q493H substitution, showed an enhanced binding affinity to hACE2. However, the third substitution of K417N significantly reduced binding activity of RBD$_{MACSp36}$ to hACE2, rendering a lower affinity than that of RBD$_{WT}$. Collectively, these results demonstrated that MASCp36 retained the high binding affinity to hACE2.

Furthermore, to confirm whether the mouse-adapted strain MASCp6 retained the infectivity to human cells, primary human airway epithelia (HAE) culture was infected with MASCp36 and WT SARS-CoV-2, respectively. Although both MASCp36 retained the infectivity to HAE (Fig. 9a), the progeny viral RNA loads of MASCp36 were much lower than that of WT SARS-CoV-2 (Fig. 9b), which was consistent to the binding affinity assay.

**Structural basis for the enhanced virulence of MASCp36.** Finally, to elucidate the molecular basis for the gradual changes in

specificity of MASCp36, structural investigations of the mACE2 in complex with RBD$_{MASCp25}$ or RBD$_{MASCp36}$ were carried out. Two non-competing Fab fragments that recognize the RBD beyond the mACE2 binding sites were used to increase the molecular weight of this complex for pursuing an atomic resolution by cryo-EM reconstruction (Supplementary Figs. 5–8). Interestingly, cryo-EM characterization of the mACE2 in complex with RBD$_{MASCp25}$ revealed that the complex adopts three distinct conformational states, corresponding to tight binding (state 1), loose binding (state 2), and no binding modes (state 3) (Supplementary Fig. 9), indicative of a quick association and quick dissociation interaction manner between the mACE2 and RBD$_{MASCp25}$. However, only the tight binding conformation was observed in the mACE2-RBD$_{MASCp36}$ complex structure, reflecting a more stable/mature binding mode for the RBD$_{MASCp36}$ to mACE2, akin to that of the RBD$_{WT}$ and hACE2. We determined asymmetric cryo-EM reconstructions of the mACE2-RBD$_{MASCp36}$ complex at 3.7 Å and three states of the mACE2-RBD$_{MASCp25}$ complex at 4.4–8.2 Å (Supplementary Figs. 6, 7, 9, and 10 and Table 1, Supplementary Table 1). The map quality around the mACE2-RBD$_{MASCp36}$ interface was of sufficient quality for a reliable analysis of the interactions (Fig. 10a, Supplementary Fig. 7, and Supplementary Table 2).

The overall structure of the mACE2-RBD$_{MASCp36}$ complex resembles that of the RBD$_{WT}$-hACE2 complex with a root mean square deviation of 1.0 Å (Supplementary Fig. 8). The RBD$_{MASCp36}$ recognizes the helices (α1 and α2) located at the apical region of the mACE2 via its receptor-binding motif (RBM) (Fig. 10a–c). The interaction area on the mACE2 could be primarily divided into three patches (PI, PII, and PIII), involving extensive hydrophilic and hydrophobic interactions with three regions separately clustered by three adaptation-mediated mutated residues (K417N, corresponding to Clus1; Q493H, corresponding to Clus2; and N501Y, corresponding Clus3) in the RBM (Fig. 10c–e). Coincidentally, a number of amino acid substitutions, such as Q493K, Q498Y, and P499T, in the RBM identified in other reported mouse-adapted SARS-CoV-2 isolates[14,15,22] were included either in the Clus2 or Clus3,

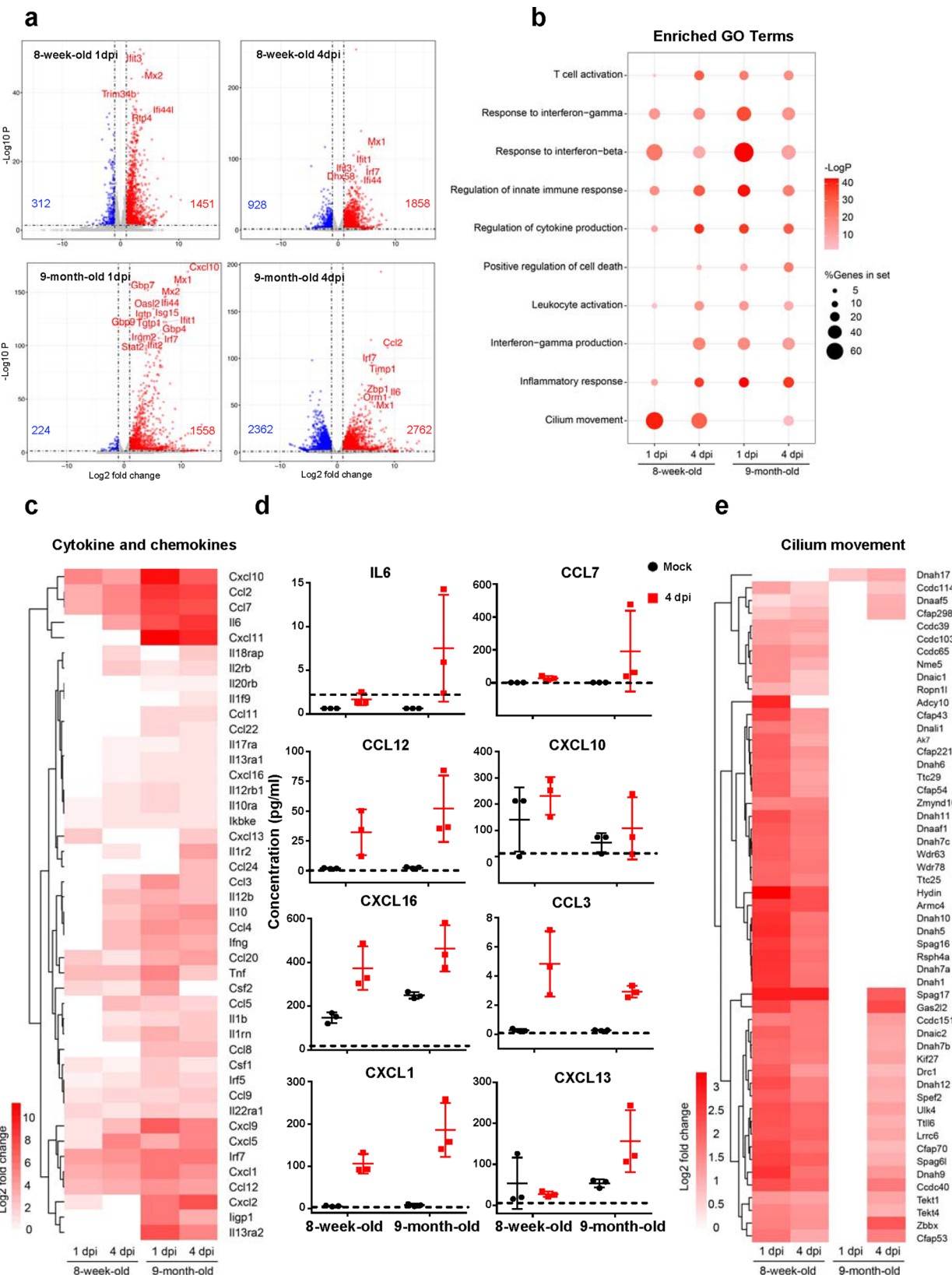

underlining the putative determinants for cross-transmission (Fig. 10d, e). An extra Clus1 is further accumulated in the MASCp36 to gain utmost binding activity and infection efficacy (Fig. 10d, e). The extensive hydrophobic interactions in Clus3 constructed by Y501 (or Y498 or H498 in other mouse-adapted SARS-CoV-2 isolates), Y505 in the RBD$_{MASCp36}$ and Y41, H353

in the mACE2, hydrogen bonds in Clus2 formed H493 (K493 in other mouse-adapted strain) in the RBD$_{MASCp36}$ and N31, E35 in the mACE2 and hydrophilic contacts constituted by N417 in the RBD$_{MASCp36}$ and N30, Q34 in the mACE2 contribute to the tight binding of the MASCp36 to mACE2. Contrarily, structural superimposition of the RBD$_{WT}$ over the mACE2-RBD$_{MASCp36}$

**Fig. 5 Host transcriptional response to MASCp36 in lungs of 8-week-old and 9-month-old male BALB/c mice. a** Volcano plots indicating differential regulated genes after MASCp36 infection at 1 and 4 dpi. Upregulated genes ($P < 0.05$) with a log2 (fold change) of more than 1 are indicated in red, downregulated genes ($P < 0.05$) with a log2 (fold change) of less than –1 are indicated in blue. Among the top 20 upregulated genes, interferon stimulated genes (ISGs) and cytokines were marked with gene symbols. DESeq2 that uses Wald test was used to identify differentially regulated genes between mock and infected groups ($n = 3$ per group). $P$ indicate Benjamini–Hochberg-adjusted $P$ values. **b** Dot plot visualization of enriched GO terms of upregulated genes at 1 and 4 dpi. Gene enrichment analyses were performed using Metascape against the GO dataset for biological processes. $P$ values were calculated based on accumulative hypergeometric distribution. The color of the dots represents the –Log$P$ value for each enriched GO term, and size represents the percentage of genes enriched in each GO term. **c** Heatmap indicating the expression patterns of genes belonging to GO annotation for cilium movement. **d** Male BALB/c mice (8-week-old and 9-month-old) were i.n. inoculated with 12,000 PFU of MASCp36, and lung homogenates were prepared at 4 dpi ($n = 3$ per group). Cytokine and chemokine analysis was determined by Luminex. Dash lines denote the detection limit. **e** Heatmap indicating the expression patterns of 44 cytokine and chemokine genes. **c** and **d** depict the log2(fold change) of genes of infected compared with mock-infected mice. The log2(fold change) of not significantly changed genes ($P > 0.05$) were counted as zero.

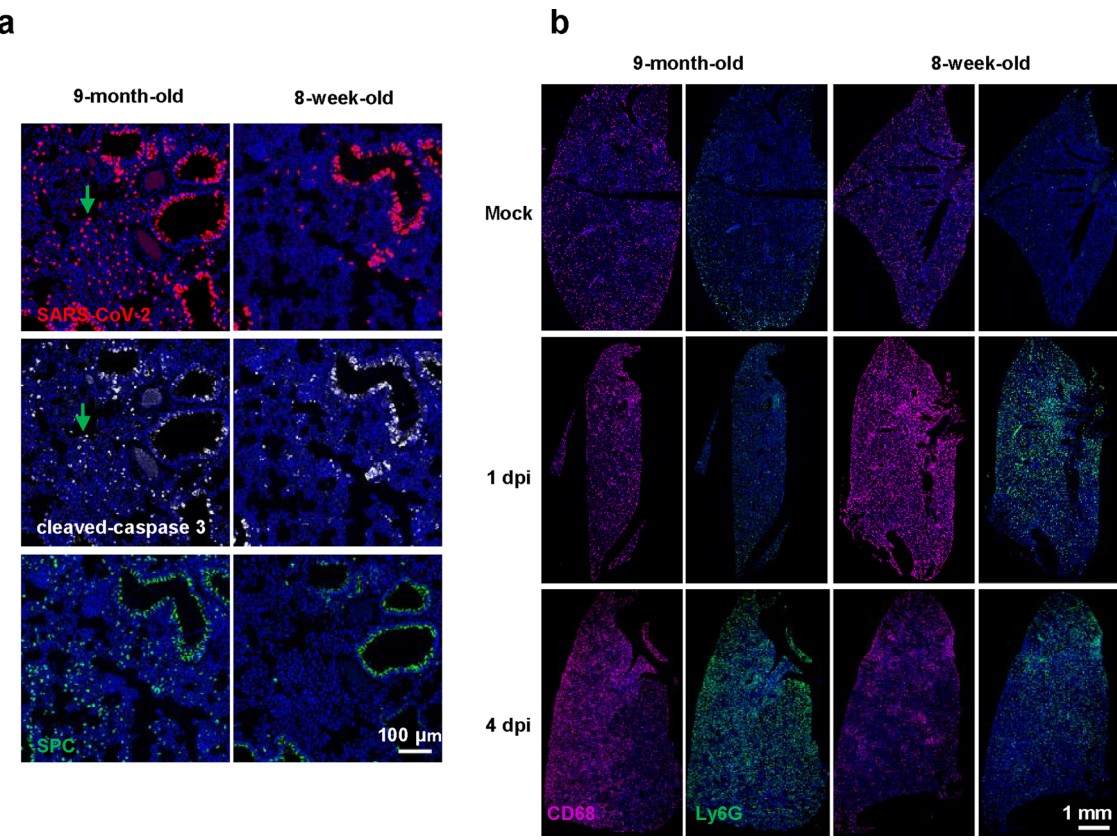

**Fig. 6 Lung-specific response of 8-week-old and 9-month-old male BALB/c mice during MASCp36 infection. a** Multiplex immunofluorescence staining for detection of SARS-CoV-2 N protein (red), Cleaved caspase-3 (white), and SPC (green) expression in lung tissues of male mice (9-month-old and 8-week-old) at 1 dpi. The green arrows indicate apoptotic cells with MASCp36 infection in the alveolar region ($n = 3$ per group) **b** Multiplex immunofluorescence staining for CD68[+] macrophages and Ly-6G[+] neutrophils infiltration in 9-month-old mice and 8-week-old mice after infection at 1 and 4 dpi. Less CD68[+] macrophages and Ly-6G[+] neutrophils were detected in 9-month-old mice than that in young mice 1 dpi, and reversed on 4 dpi ($n = 3$ per group).

complex reveals the loss of these interactions, leading to the inability of the RBD$_{WT}$ to bind mACE2 (Fig. 9e).

Meanwhile, the atomic structures of hACE2 in complex with RBD$_{MACSp6}$, RBD$_{MACSp25}$, or RBD$_{MACSp36}$ were also solved at 3.1–3.7 Å (Supplementary Fig. 11 and Table 1). The same as RBD$_{MACSp36}$−mACE2 complex, the interaction region could be divided into three patches, the three adaption-mutated residues (K417N, located at PI; Q493H, located at PII; and N501Y, located at PIII) were located at the three patches, respectively (Fig. 10f). The extensive hydrophobic interactions in PIII were constructed by Y501, T500 in the RBD and Y41, L45 in the hACE2, salt bridge interactions in PII were formed by H493 in the RBD and E35 in

the hACE2. However, in PI, the substitution of K with N at residue 417 in RBD$_{MACSp36}$ loses the salt bridge interaction, which would be formed by K417 in the RBD$_{WT}$ and D30 in the hACE2, structurally explaining the reduced binding affinity of RBD$_{MACSp36}$ to hACE2. Overall, these analysis pinpoints key structure-infectivity correlates, unveiling the molecular basis for host adaptation-mediated evolution of SARS-CoV-2.

## Discussion

Clinically, the severe COVID-19 disease onset might result in death due to massive alveolar damage and progressive respiratory failure[23,24]. However, most animal models previously described

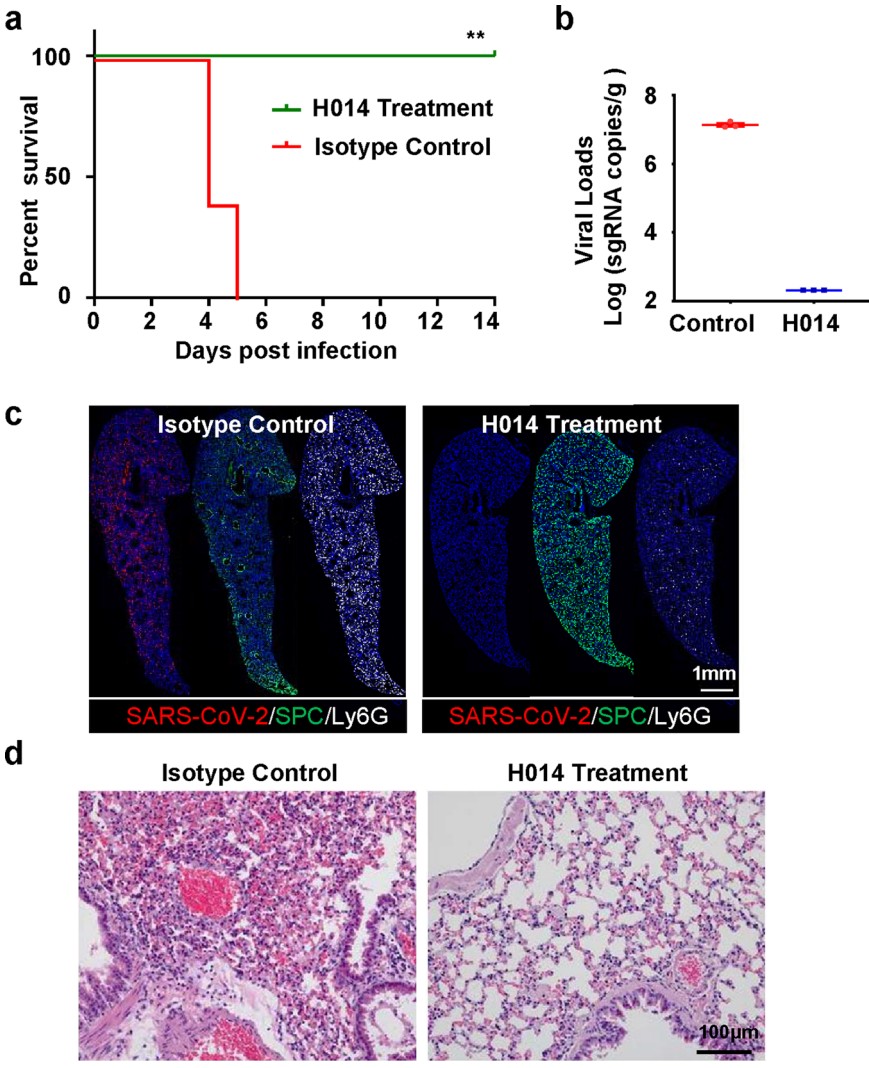

**Fig. 7 The efficacy of H014 against MASCp36 infection.** Male BALB/c mice (9-month-old) infected with 1200 PFU of MASCp36. **a** Survival curves of mice in H014 and mock treatment groups ($n = 5$ per group). Statistical significance was analyzed by Log-rank (Mantel–Cox) test ($P = 0.0023$, $**P < 0.01$). **b** Lung samples were collected at 4 dpi and subjected to viral RNA load analysis by RT-qPCR. The result were presented as mean ± SD. Dash lines denote the detection limit ($n = 3$ per group). **c** Multiplex immunofluorescence staining of mouse lung sections for detection of SARS-CoV-2 N protein (red), SPC (green), and Ly6G (white) ($n = 3$ per group). **d** H&E staining of lung sections from H014 or isotype control ($n = 3$ per group).

recaptulated the mild to moderate clinical symptoms of COVID-19. The MASCp36-based mouse model described here recapitulated most spectrums of seriously ill COVID-19 patients caused by SARS-CoV-2 infection, such as pulmonary oedema, fibrin plugs in alveolar, hyaline membrane, and scattered hemorrhage[25,26]. The complicated immunopathological phenomena observed in severe COVID-19 patients, such as massive macrophages and neutrophils infiltration, and excessively increased proinflammatory cytokines such as IL-6, were also observed in this MASCp36-infected mouse model. Herein, 9-month-old, male BALB/c mice infected with MASCp36 developed DAD, as well as comprehensive vasculature damage (Fig. 3). Importantly, thick fluid in pericardial cavity, hemorrhage, and severe edema with less lymphocyte-cuff were observed in lung tissue, resembling clinical manifestations of severe COVID-19[4].

Lung fibrosis and regeneration were also observed in this model that is the signs of fibrosis in clinic. Although SARS-CoV-2 viral antigen has been detected in kidney of postmortem specimens[27], no viral antigen or viral RNA were detected in our model (Fig. 1e, f). So in this MASCp36-infected mouse model, the kidney injury may arise due to secondary endothelial injury

leading to proteinuria[28]. In addition, although SARS-CoV-2 has also been implicated to have neurotropic potential in COVID-19[29], we did not find typical characteristics of viral encephalitis in this model. Importantly, the imbalanced immune response with high levels of proinflammatory cytokines, increased neutrophils and decreased lymphocytes, which were in line with SARS-CoV and MERS-CoV infections[30], playing a major role in the pathogenesis of COVID-19[31], were also observed in this model.

The skewed age distribution of COVID-19 disease was reproduced in the MASCp36-infected mouse model where more severe symptoms were observed in 9-month-old mice when compared to young mice. Different from H1N1 pandemic[32], COVID-19 appears to have a mild effect on populations under 30 years, and the elderly are more likely to progress to severe disease and are admitted to intensive care unit worldwide[33]. ACE2, the functional receptor of SARS-CoV-2, expressed increasingly in the lungs with age, which might provide an explanation to the higher disease severity observed in older patients with COVID-19[34]. More importantly, the host immune response may determine the outcome of the disease. Our immune system is composed of innate immunity and adaptive immunity. The innate immunity

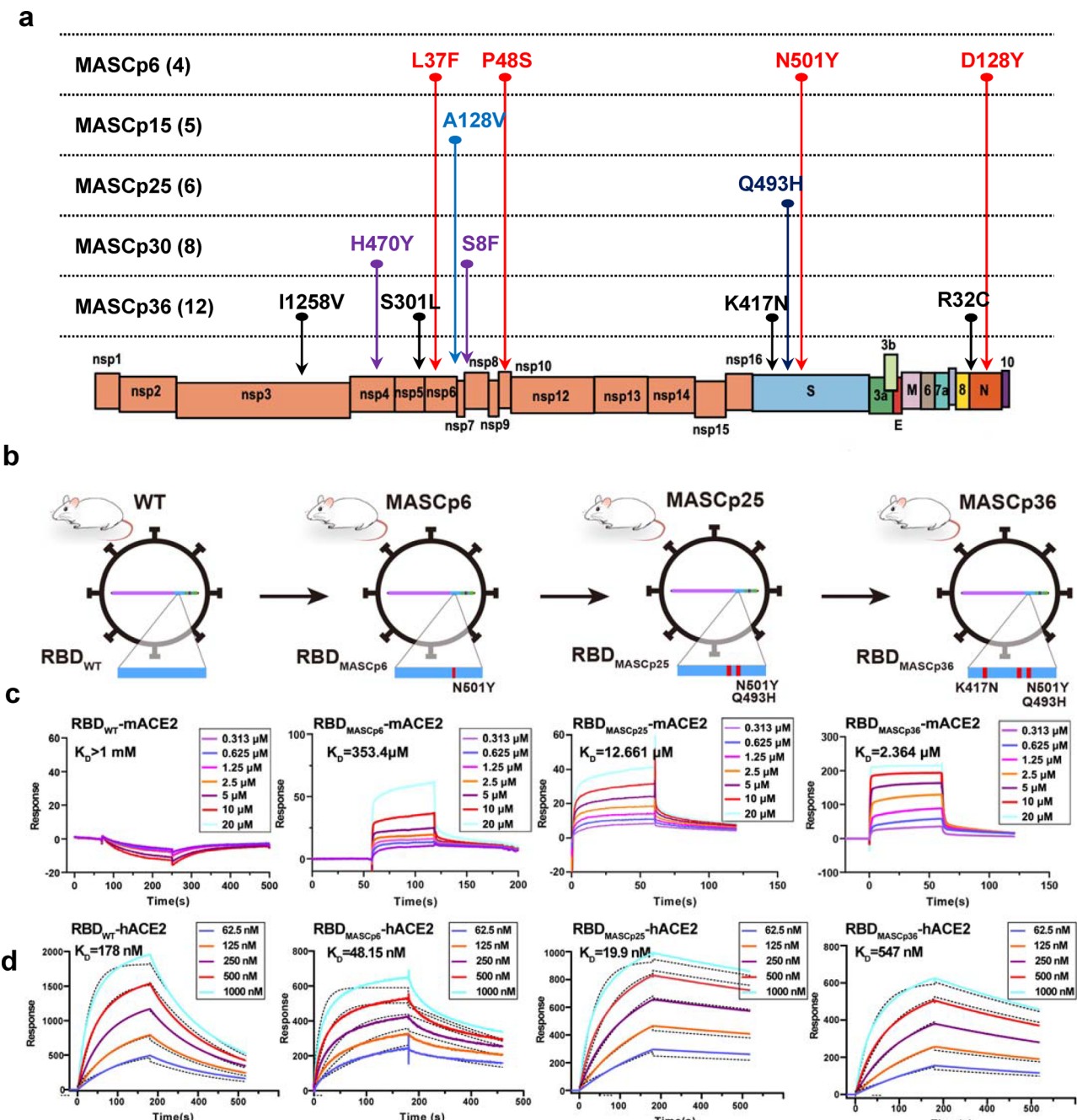

**Fig. 8 Adaptive RBD mutations identified in different mouse-adapted strains. a** Schematic diagram of SARS-CoV-2 genome and all the adaptive mutations. **b** Schematic diagram depicting virus harboring different RBD mutations. The mutation site is marked with a red rectangle. The proportion of N501Y, Q493H, K417N mutations located on the RBD. **c, d** Binding properties of $RBD_{WT}$, $RBD_{MASCp6}$, $RBD_{MASCp25}$, and $RBD_{MASCp36}$ to mACE2 and hACE2 analyzed using SPR. For both panels, mACE2 was loaded onto the sensor; $RBD_{WT}$, $RBD_{MASCp6}$, $RBD_{MASCp25}$, and $RBD_{MASCp36}$ were injected. Response units were plotted against protein concentrations. The KD values were calculated by BIAcore® 3000 analysis software (BIAevaluation version 4.1).

comprises of the first line of defense against pathogens and is acute as well as short lived. However, aging is linked with insufficient, prolonged, and chronic activation of innate immunity associated with low-grade and systemic increases in inflammation (inflamm-aging) that can be detrimental for the body[35]. The delicate co-operation and balance are interrupted by the chronic activation of innate immunity and declined adaptive immune responses with increasing age in COVID-19[36]. In the MASCp36-infected mouse model, the young mice presented acute inflammatory response with more innate immune cells

infiltration on day 1, while lagged and sustained immune response in 9-month-old mice. After further analyzing the RNA-Seq results, we observed robust innate immune response at 1 dpi in 9-month-old mice. While we found that T cell activation in 9-month-old mice was not as strong as in 8-week-old mice at 4 dpi. Meanwhile, we found that genes involved in "cilium movement" were significantly stimulated at 1 and 4 dpi in 8-week-old male mice, while much less genes related to cilium movement were upregulated in 9-month-old male mice till 4 dpi. Previous studies show that mucociliary clearance is an important defense

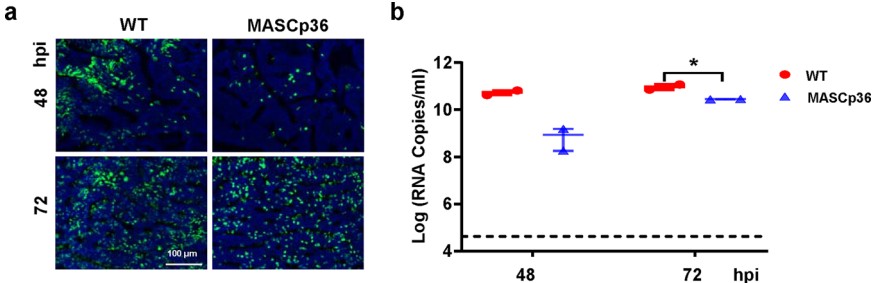

**Fig. 9 Characterization of MASCp6 and MASCp36 in human airway epithelial (HAE) cells. a** Immunofluorescence staining of WT and MASCp36-infected HAE cultures for SARS-CoV-2 N protein (green) and DAPI (blue) at 48 and 72 h post infection. The percentage of SARS-CoV-2 N protein positive cells is presented as mean ± SD (n = 4). **b** Growth kinetics of WT (BetaCoV/Beijing/IMEBJ05/2020, Nos. GWHACBB01000000) and MASCp36 in HAE at 48 and 72 h post infection. Results are shown as mean ± SD from two independent replicates. The exact P values of 48 and 72 h post infection were 0.0905 and 0.0455 analyzed by two-way ANOVA.

**Table 1 Cryo-EM data collection and atomic model refinement statistics.**

| Protein | RBD$_{MACSp6}$-Fab$_{B8}$-Fab$_{D14}$-hACE2 | RBD$_{MACSp25}$-Fab$_{B8}$-Fab$_{D14}$-hACE2 | RBD$_{MACSp36}$-Fab$_{B8}$-Fab$_{D14}$-hACE2 | RBD$_{MACSp36}$-Fab$_{B8}$-Fab$_{D14}$-mACE2 |
|---|---|---|---|---|
| PDB code | 7FDG | 7FDH | 7FDI | 7FDK |
| *Data collection and reconstruction statistics* | | | | |
| 0141Voltage (kV) | 300 | 300 | 300 | 300 |
| Detector | K2 | K2 | K2 | K2 |
| Pixel size (Å) | 1.04 | 1.04 | 1.04 | 1.04 |
| Electron dose (e⁻/Å²) | 60 | 60 | 60 | 60 |
| Defocus range (µm) | 1.25–2.7 | 1.25–2.7 | 1.25–2.7 | 1.25–2.7 |
| Final particles | 69,324 | 68,211 | 85,978 | 162,615 |
| Resolution (Å) | 3.78 | 3.72 | 3.12 | 3.69 |
| *Models refinement and validation statistics* | | | | |
| Ramachandran statistics | | | | |
| Favored (%) | 94.65 | 94.9 | 96.7 | 92.38 |
| Allowed (%) | 5.34 | 4.99 | 3.30 | 7.43 |
| Outliers (%) | 0.00 | 0.00 | 0.00 | 0.11 |
| Rotamer outliers (%) | 0.01 | 0.03 | 0.00 | 0.09 |
| R.m.s.d | | | | |
| Bond lengths (Å) | 0.01 | 0.02 | 0.01 | 0.02 |
| Bond angles (°) | 1.07 | 1.23 | 1.02 | 1.26 |

mechanism in the respiratory tract that requires coordinated ciliary activity and proper mucus production to propel airway surface liquids that traps pathogens and pollutants, permitting their clearance from the lungs[37–39]. A recent study on ACE2 shows the receptor protein robustly localizes within the motile cilia of airway epithelial cells, which likely represents the initial or early subcellular site of SARS-CoV-2 viral entry during host respiratory transmission[40]. So, we concluded that stimulation of cilium movement may benefit young mice to clear virus at the early stage of infection, and over-stimulated innate immune response and reduced adaptive immune response may lead to higher mortality in male 9-month-old mice. The different immune response in mice model may be vital in limiting virus replication at early times and contribute to different outcome on day 4 in young or 9-month-old mice.

In addition to the age-related skewed distribution of COVID-19, gender-related differences in distribution of COVID-19 disease are also recapitulated in this MASCp36-infected mouse model with increased susceptibility and enhanced pathogenicity observed in male mice when compared to their female counterparts. Biological sex is an important determinant of COVID-19 disease severity[41]. In China, the death rate among confirmed cases is 2.8% for women and 4.7% for men[34]. In Italy, half of the confirmed COVID-19 cases are men that account for 65% of all

deaths[42]. This pattern is generally consistent around the world. The skewed distribution of COVID-19 suggests that physiological differences between male and female may cause differential response to infection. So the hypothesis that females display reduced susceptibility to viral infections may be due to the stronger immune responses they mount than males[43]. In this study, based on the RNA-Seq results, we also found that female 9-month-old mice show higher intrinsic expression level of immune response-related genes than male 9-month-old mice. Stronger immune response was also observed in female 9-month-old mice after infection. It has been studied that androgens may lower and estrogens may enhance several aspects of host immunity. Channappanavar et al.[44] studied that estrogen receptor signaling is critical for protection in females against SARS-CoV infection. So, we speculated here that estrogen may also play an important role in the protection against MASCp36 infection in female mouse model. In addition, androgens facilitate and estrogens suppress lymphocyte apoptosis. Furthermore, genes on the X chromosome important for regulating immune functions, and androgens may suppress the expression of disease resistance genes such as the immunoglobulin superfamily[45]. In the MASCp36-infected mouse model, we found out that it presented higher mortality of the male than the female infected with the same dose of virus, indicating the successful recapitulation of COVID-19 and also its

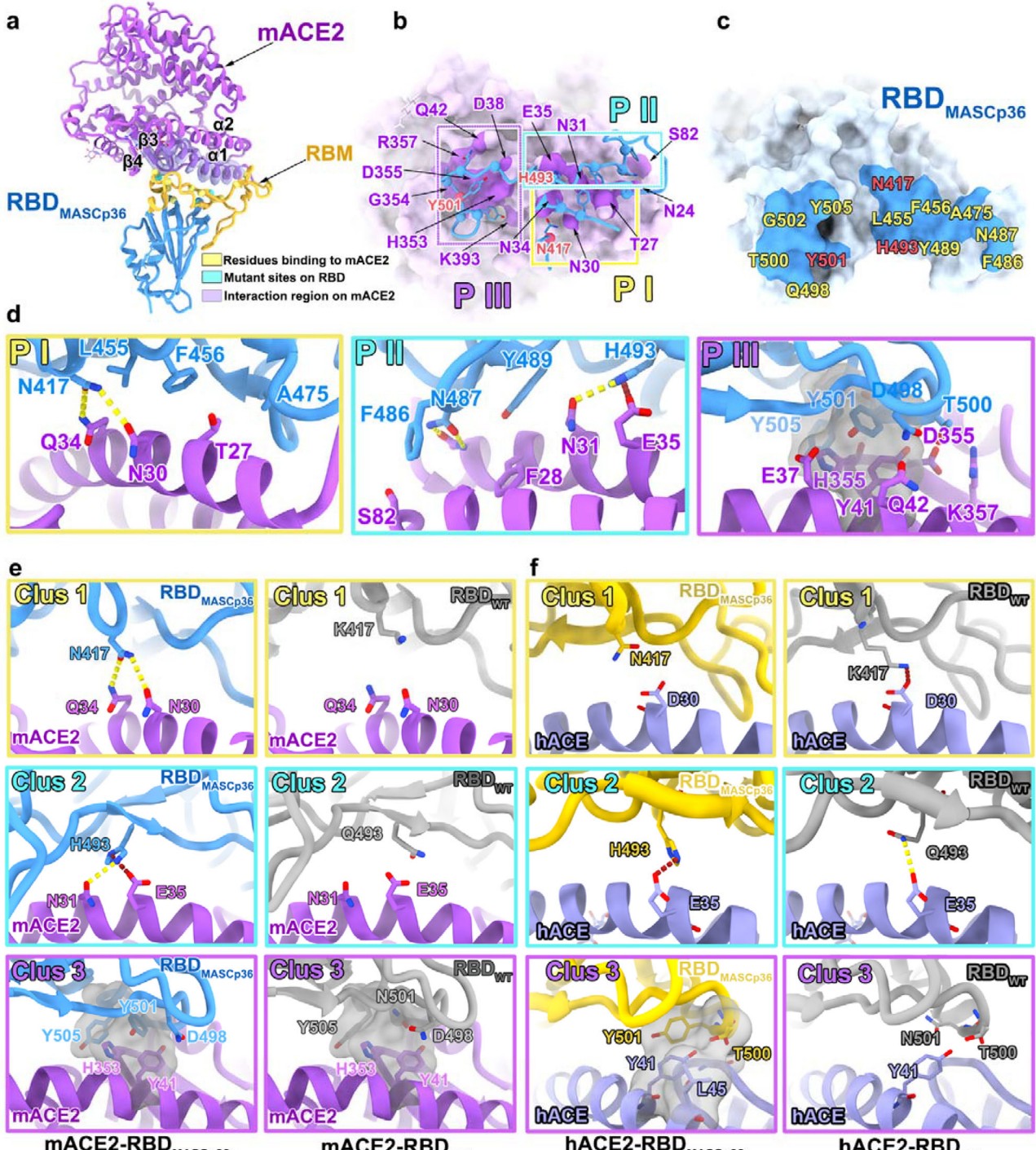

**Fig. 10 Mechanism of binding of SARS-CoV-2-derived MASCp36 to mACE2. a** Overall structure of RBD$_{MASCp36}$ bound to mACE2. Residues of RBD$_{MASCp36}$ participating in the binding to mACE2 are shown as spheres and colored in yellow, the mutation sites are colored in cyan, the RBM is colored in gold. The region of mACE2 responsible for binding is labeled. **b, c** The binding interface between RBD$_{MASCp36}$ and mACE2. The residues involved in binding to mACE2 are presented as sticks, and the residues of mACE2 interacting with RBD$_{MASCp36}$ are shown as surface. The mutated residues in RBD$_{MASCp36}$ are colored in red. **d** Details of the interactions between RBD$_{MASCp36}$ and mACE2. Some residues involved in the formation of hydrophobic patches (gray mesh), salt bridge (red dash), and hydrogen bonds (yellow dash) are shown as sticks and labeled. **e** The comparison of interactions at RBD$_{WT}$-mACE2 and RBD$_{MASCp36}$-mACE2 interface. RBD$_{WT}$, RBD$_{MASCp36}$, and mACE2 are colored in gray, cyan, and purple, respectively. The residues involved in the formation of hydrophobic patches (gray mesh), salt bridge (red dash), and hydrogen bonds (yellow dash) are shown as sticks and labeled. **f** The comparison of interactions at RBD$_{WT}$-hACE2, RBD$_{MASCp6}$- hACE2, RBD$_{MASCp25}$-hACE2, and RBD$_{MASCp36}$-hACE2 interface. RBD$_{WT}$, RBD$_{MASCp6}$, RBD$_{MASCp25}$, and RBD$_{MASCp36}$ are colored in gray, red, yellow, and cyan, respectively.

potential application in the study of the pathogenesis of the disease.

A total of 12 amino acid mutations were identified in the genome of MASCp36 in comparison with WT SARS-CoV-2 strain. The sequentially acquired triple substitutions N501Y/Q493H/K417N in S protein of MASCp36 increased their affinities to mACE2, thus contributed to enhanced infectivity and lethal phenotype mice. Interestingly, MASCp36 showed decreased affinity to hACE2 (Fig. 8d) as well as decreased infectivity to primary HAE cells (Fig. 9) in comparison with WT SARS-CoV-2. These experimental observations were further verified by structural analysis in which MASCp36 loses the key salt bridge interaction with hACE2 when compared to WT SARS-CoV-2 (Fig. 10f). More importantly, Cryo-EM structures of both hACE2 and mACE2 in complex with RBD_MASCp25 and RBD_MASCp36 define preciously the atomic determinants of the receptor-binding switch: N501Y/Q493H/K417N in MASCp36 formed tight interactions with mACE2 in three patches, respectively, while K417N reduced interaction with hACE2. Thus, MASCp36 showed enhanced virulence in mice whereas decreased infectivity to human cells.

In addition, there are nine amino acid substitutions outside the S protein of MASCp36 (Fig. 8a). Notably, R32C and D128Y in the N protein deserve special attention. The D128Y mutation in N protein has been well recorded in human variants from D128Y[46]. A recent report demonstrated acts as an antagonist of IFN and viral encoded repressor of RNA interference[47]. Previous studies also reported N protein help package encapsidated genome into virions via binding Nps3[48]. Also, mutations were noted in Nsp3 and Nsp5 that play role in blocking host innate immune response, promoting cytokine expression, and inhibiting IFN signaling. Further studies will focus on the influence of virus changes to host innate immunity in mice[49]. Recently, Shi et al. found that SARS-CoV-2 nsp1 and nsp6 suppress IFN-I signaling that provides insights on viral evasion and its potential impact on viral transmission and pathogenesis[50]. Whatever, multiple amino acid substitution might contribute to the enhanced virulence phenotype, independently or synergistically. At present, we cannot rule out the contribution of these mutations, and further validation with reverse genetic tools will help understand the biological function of each single mutation[51].

There is another interesting point that deserves concerns. These adaptive mutations emerged during mouse passage; however, many of them have been documented in SARS-CoV-2 variants during human transmission. For example, the 501Y.V1 variant firstly detected in the United Kingdom contained the unique N501Y mutation in RBD[52], and more recent SARS-CoV-2 variants (501Y.V2 and 501Y.V2) contained both N501Y and K417N substitutions in S protein[53]. Except for N501Y and K417N, several other amino acid substitutions in MASCp36 were also found in human variants, including L37F in nsp6 and D128Y in N protein[46,54]. The coincidental emergence of the same mutations during mice passaging and COVID-19 epidemics highlights the importance of these mutations during host adaption.

In conclusion, our MASCp36 mouse model exhibited symptoms of DAD and ALI in both laboratory standard BALB/c and C57BL/6 rodent model, which largely simulated COVID-19 severe disease. And the age- and gender-dependent mortality well reproduced the clinical findings of human COVID-19. This model will be of high value for studying the pathogenesis of COVID-19 with genetic modified mice and for the rapid efficacy tests of potent countermeasures against SARS-CoV-2.

## Methods

**Ethics statement.** All procedures involving infectious virus were conducted in Biosafety Level 3 laboratory and approved by the Animal Experiment Committee of Laboratory Animal Center, Beijing Institute of Microbiology and Epidemiology (approval number: IACUC-DWZX-2020-002).

**Virus and mice.** Mouse-adapted strain of SARS-CoV-2 (MASCp6) was developed from a clinical SARS-CoV-2 isolate in our previous study[13]. Additional serial passage of 30 times was performed as previously described[13]. Briefly, we administered i.n. to 9-month-old BALB/c mice with dose of $1.6 \times 10^4$ PFU of MASCp6. After 3 dpi, the lungs of each mice were collected and homogenized (10% w/v) in DMEM (Invitrogen, USA). Then, the supernatant of lung homogenate was clarified by centrifugation and filtration. The other three naive mice were administered i.n. with the mixing well lung homogenate from MASCp6-infected mice. The process of intranasally (i.n.) inoculation of two to four female aged BALB/c mice was repeated 30 times. BALB/c and C57BL/6 mice were purchased from Beijing HFK Bioscience Co., Ltd. and Beijing Vitalriver Laboratory Animal Technology Co. Ltd. The virus stock of MASCp36 was amplified and titrated by standard plaque forming assay on Vero cells.

**Measurement of viral sgRNA.** Tissue homogenates were clarified by centrifugation a 3500 g for 6 min, and the supernatants were transferred to a new EP tube. RNA was extracted using the QIAamp Viral RNA Mini Kit (Giagen) according to the manufacturer's protocol[13]. sgRNA quantification in each sample was performed by quantitative reverse transcription PCR (RT-qPCR) targeting the E subgenomic mRNA of SARS-CoV-2. RT-qPCR was performed using One Step PrimeScript RT-PCR Kit (Takara, Japan) with the in primers and probes (Supplementary Table 3)[55].

**Deep sequencing.** Total RNA was extracted after each passage using High Pure Viral RNA Kit (Roche, Switzerland), and the purified viral RNA was used for library construction using Ion Total RNA-Seq kit V2 (Thermo Fisher, USA). The library was sequenced on an Ion Torrent S5Plus sequencer (Thermo Fisher, USA). Sequences were assembled and analyzed with CLC Genomic Workbench (Qiagen, Germany). All reads were mapped to SARS-CoV-2 reference genome (Wuhan-Hu-1, GenBank accession number MN908947). The consensus sequence was extracted. Single-nucleotide variations with variation proportion above 1% were called using this software.

**Mouse virulence study.** Six weeks and 9 months BALB/c and C57BL/6 female and male mice were maintained in a pathogen-free facility and housed in cages containing sterilized feed and drinking water. Following intraperitoneal anesthetization with sodium pentobarbital, mice were i.n. inoculated with varying doses of MASCp36 or the same volume of phosphate-buffered saline (PBS) for mock infection. Four mice of each group were sacrificed on days 1 and 4 after infection for lung damage study and sgRNA quantification. Clinical manifestation and survival were recorded for 14 days.

**Histopathological analysis.** Lung, heart, liver, spleen, kidney, brain, intestine, and trachea tissues were collected 4 dpi with MASCp36, and paraffin-embedded in accordance with standard procedure. Sections at 4-μm thickness were stained with H&E and examined by light microscopy and analyzed by two experienced experimental pathologists.

**Multiplex immunofluorescent assay.** The multiplex immunofluorescence assay was conducted as previously described[13]. Briefly, the retrieved sections were incubated with primary antibody for 2 h followed by detection using the HRP-conjugated secondary antibody and TSA-dendron-fluorophores (NEON 7-color Allround Discovery Kit for FFPE, Histova Biotechnology, NEFP750). Afterwards, the primary and secondary antibodies were thoroughly eliminated by heating the slides in retrieval/elution buffer (Abcracker®, Histova Biotechnology, ABCFR5L) for 10 s at 95 °C using microwave. In a serial fashion, each antigen was labeled by distinct fluorophores. Multiplex antibody panels applied in this study include: ACE2 (Abcam, ab108252, 1:200); SARS-CoV-2 N protein (Sinobiological, 40143-R004, 1:2000); CC10 (Millipore, 07-623, 1:500), FOXJ1 (Abcam, ab235445, 1:1000), SPC (Abcam, ab211326, 1:500); Cleaved caspase-3 (CST, 9664, 1:300); Ly-6G (CST, 87048, 1:400); CD68 (CST, 97778, 1:300); CD3 (CST, 78588, 1:300); CD19 (Abcam, ab245235, 1:800); ICOS (Abcam, ab224644, 1:200); CD31 (CST, 77699, 1:300). After all the antibodies were detected sequentially, the slides were imaged using the confocal laser scanning microscopy platform Zeiss LSM880. Some data were further processed and statistically analyzed using Bitplane Imaris software (Bitplane AG, Zurich, Switzerland).

**Cytokine and chemokines analysis.** Cytokines and chemokines in mouse lung homogenates were measured using a Bio-Plex Pro Mouse Cytokine Grp I Panel 31-Plex (Bio-Rad, USA) according to the manufacturer's protocol. The data were collected on Luminex 200 and analyzed by Luminex PONENT (Thermo Fisher, USA).

**RNA sequencing and bioinformatic analyses.** Lung tissue (~1 mg) from MASCp36-infected mice (male, 8-week-old; male, 9-month-old; female, 9-month-old) or mock treated mice were collected into RNase-free Grinding tube and 1 mL of TRIzol was added. And then the lung tissues were homogenized at 4 °C, 75 HZ, 240 s. Total RNA from lung tissues were extracted using TRIzol (Invitrogen, USA) and treated with DNase I (NEB, USA). Sequencing libraries were generated using NEBNext® UltraTM RNA Library Prep Kit for Illumina® (NEB, USA) following the manufacturer's recommendations and index codes were added to attribute sequences to each sample. The clustering of the index-coded samples was performed on a cBot cluster generation system using HiSeq PE Cluster Kit v4-cBot-HS (Illumina) according to the manufacturer's instructions. After cluster generation, the libraries were sequenced on Illumina Novaseq6000 platform and 150 bp paired-end reads were generated. After sequencing, perl script was used to filter the original data (Raw Data) to clean reads by removing contaminated reads for adapters and low-quality reads. Clean reads were aligned to the mouse genome (Mus_musculus GRCm38.99) using Hisat2 v2.1.0. The number of reads mapped to each gene in each sample was counted by HTSeq v0.6.0 and TPM (Transcripts Per Kilobase of exon model per Million mapped reads) was then calculated to estimate the expression level of genes in each sample. DESeq2 v1.6.3 was used for differential gene expression analysis. Genes with $Padj \leq 0.05$ and $|Log2FC| > 1$ were identified as DEGs. DEGs were used as query to search for enriched biological processes (Gene ontology BP) using Metascape. Heatmaps of gene expression levels were constructed using heatmap package in R (https://cran.rstudio.com/web/packages/pheatmap/index.html). Dot plots and volcano plots were constructed using ggplot2 (https://ggplot2.tidyverse.org/) package in R.

**In vivo efficacy assay with MASCp36 model.** Group of 9-month male BALB/c mice was inoculated i.n. with 40 μL MASCp36 (1200 PFU) and intraperitoneally administered 200 μL of H014 (50 mg/kg) at 24 h before and after MASCp36 infection. The same volume of an isotype antibody was administrated as control. On day 4 post infection, three mice in each group were sacrificed and lung tissues were prepared for pathology analysis and sgRNA quantification. The other six mice in each group were monitored for survival for 14 days.

**Protein expression and purification.** The cloning and production of SARS-CoV-2 RBD (residues 319-541, GenBank: MN_908947.3), RBD mutants (RBD$_{MACSp25}$: Q493H, N501Y; RBD$_{MACSp36}$: K417N, Q493H, N501Y), hACE2 (residues 19-624, GenBank: NM_021804.3), and mACE2 (residues 19-739, GenBank: NM_001130513.1) were synthesized and subcloned into the mammalian expression vector pCAGGS with a C-terminal 2 × StrepTag to facilitate protein purification. Briefly, RBD$_{WT}$, RBD$_{MACSp25}$, RBD$_{MACSp36}$, hACE2, and mACE2 were expressed by transient transfection of HEK Expi 293F cells (Gibco, Thermo Fisher, A14527) using Polyethylenimine Max Mw 40,000 (Polysciences). The target protein was purified from clarified cell supernatants 3 days post transfection using StrepTactin resin (IBA). The resulting protein samples were further purified by size-exclusion chromatography using a Superdex 75 10/300 column (GE Healthcare) or a Superdex 200 10/300 Increase column (GE Healthcare) in 20 mM HEPES, 200 mM NaCl, pH 7.0.

To purify the final quaternary complex (RBD$_{MACSp25}$/RBD$_{MACSp36}$-Fab$_{B8}$-Fab$_{D14}$-mACE2), firstly, the ternary complex (RBD$_{MACSp25}$/RBD$_{MACSp36}$ -Fab$_{B8}$-Fab$_{D14}$) was assembled. RBD$_{MACSp25}$ or RBD$_{MACSp36}$ was mixed with Fab$_{B8}$ and Fab$_{D14}$ at the ratio of 1:1.2:1.2, incubated for 30 min on ice. The mixture was then subjected to gel filtration chromatography. Fractions containing the ternary complex (RBD$_{MACSp25}$/RBD$_{MACSp36}$-Fab$_{B8}$-Fab$_{D14}$) were pooled and concentrated. Then mACE2 was mixed with the ternary complex (RBD$_{MACSp25}$/RBD$_{MACSp36}$-Fab$_{B8}$-Fab$_{D14}$) at the ratio of 1:1.2 and incubated for 60 min on ice. The mixture was then subjected to Superdex 200 10/300 column (GE Healthcare). Fractions containing the quaternary complex (RBD$_{MACSp25}$/RBD$_{MACSp36}$-Fab$_{B8}$-Fab$_{D14}$-mACE2) were concentrated to 2 mg/mL. The preparation method of the RBD$_{MACSp6}$/RBD$_{MACSp25}$/RBD$_{MACSp36}$-Fab$_{B8}$-Fab$_{D14}$-hACE2 complex is the same as that of the RBD$_{MACSp25}$/RBD$_{MACSp36}$-Fab$_{B8}$-Fab$_{D14}$-mACE2 complex.

**Production of Fab fragment.** The B8 and D14 Fab fragments[56] were generated using a Pierce FAB preparation Kit (Thermo Scientific). Briefly, the antibody was mixed with immobilized-papain and then digested at 37 °C for 3–4 h. The Fab was separated from the Fc fragment and undigested IgGs by protein A affinity column and then concentrated for analysis.

**Surface plasmon resonance.** mACE2 or hACE2 was immobilized onto a CM5 sensor chip surface using the NHS/EDC method to a level of ~600 response units (RUs) using BIAcore® 3000 (GE Healthcare) and PBS as running buffer (supplemented with 0.05% Tween-20). wtRBD, RBD$_{MACSp6}$, RBD$_{MACSp25}$, and RBD$_{MACSp36}$, which were purified and diluted, and injected in concentration from high to low. The binding responses were measured; they were regenerated with 10 mM Glycine, pH 1.5 (GE Healthcare). The apparent binding affinity ($K_D$) for individual antibody was calculated using BIAcore® 3000 Evaluation Software (GE Healthcare).

For the competitive binding assays, the first sample flew over the chip at a rate of 20 uL/min for 120 s, then the second sample was injected at the same rate for another 120 s. All antibodies were evaluated at saturation concentration of 500 nM, mACE2 was at 1000 nM. All proteins were regenerated with 10 mM Glycine, pH 1.5 (GE Healthcare). The RUs were recorded at room temperature and analyzed using the same software as mentioned above.

**Cryo-EM sample preparation and data collection.** For Cryo-EM sample preparation, the quaternary complex (RBD$_{MACSp25}$/RBD$_{MACSp36}$-Fab$_{B8}$-Fab$_{D14}$-mACE2 and RBD$_{MACSp6}$/RBD$_{MACSp25}$/RBD$_{MACSp36}$-Fab$_{B8}$-Fab$_{D14}$-hACE2 complex) was diluted to 0.8 mg/mL. Holy-carbon gold grid (Quantifoil R0.6/1.0 mesh 300) was freshly glow-discharged with a Solarus 950 plasma cleaner (Gatan) for 60 s. A 3 μL aliquot of the mixture complex was transferred onto the grids, blotted with filter paper at 22 °C and 100% humidity, and plunged into the ethane using a Vitrobot Mark IV (FEI). For these complex, micrographs were collected at 300 kV using a Titan Krios microscope (Thermo Fisher), equipped with a K2 detector (Gatan, Pleasanton, CA), using SerialEM automated data collection software[56]. Movies (32 frames, each 0.2 s, total dose 60 e$^-$Å$^{-2}$) were recorded at final pixel size of 1.04 Å with a defocus of between –1.25 and –2.7 μm.

**Image processing.** For RBD$_{MACSp25}$-Fab$_{B8}$-Fab$_{D14}$-mACE2 complex, a total of 2109 micrographs were recorded. For RBD$_{MACSp36}$-Fab$_{B8}$-Fab$_{D14}$-mACE2 complex, a total of 2982 micrographs were recorded. For RBD$_{MACSp6}$/RBD$_{MACSp36}$-Fab$_{B8}$-Fab$_{D14}$-hACE2 complex, a total of 2005, 2341, and 2659 micrographs were recorded, respectively. Both sets of the data were processed in the same way. Firstly, the raw data were processed by MotionCor2, which were aligned and averaged into motion-corrected summed images. Then, the defocus value for each micrograph was determined using Gctf. Next particles were picked and extracted for two-dimensional alignment. The partial well-defined particles were selected for initial model reconstruction in Relion. The initial model was used as a reference for three-dimensional classification. After the refinement and post-processing, the overall resolution of RBD$_{MACSp36}$-Fab$_{B8}$-Fab$_{D14}$-mACE2 complex was up to 3.69 Å, on the basis of the gold-standard Fourier shell correlation (threshold = 0.143)[57]. For RBD$_{MACSp25}$-Fab$_{B8}$-Fab$_{D14}$-mACE2 complex, the ClassI complex was up to 7.89 Å, ClassII complex was up to 8.17 Å, and ClassIII complex was up to 4.4 Å. The final resolution of RBD$_{MACSp6}$-Fab$_{B8}$-Fab$_{D14}$-hACE2, RBD$_{MACSp25}$-Fab$_{B8}$-Fab$_{D14}$-hACE2, and RBD$_{MACSp36}$-Fab$_{B8}$-Fab$_{D14}$-hACE2 complex was up to 3.72, 3.76, and 3.12 Å, respectively. The quality of the local resolution was evaluated by ResMap[58].

**Model building and refinement.** The RBD$_{WT}$-hACE2 (PDB ID: 6M0J) structures was manually docked into the refined maps of RBD$_{MACSp36}$-Fab$_{B8}$-Fab$_{D14}$-mACE2, RBD$_{MACSp6}$-Fab$_{B8}$-Fab$_{D14}$-hACE2, RBD$_{MACSp25}$-Fab$_{B8}$-Fab$_{D14}$-hACE2, and RBD$_{MACSp36}$-Fab$_{B8}$-Fab$_{D14}$-hACE2 complex using UCSF Chimera[59] and further corrected manually by real-space refinement in COOT[60]. The atomic models were further refined by positional and B-factor refinement in real space using Phenix[61]. Validation of the final model was performed with Molprobity[59]. The data sets and refinement statistics are shown in Table 1 and Supplementary Table 1.

**Infectivity assay of MASCp36 in HAE model.** HAE was gifted from Professor Tan Lab[62]. HAE cultures were generated in an air-liquid interface for 4–6 weeks to form well-differentiated, polarized cultures that resemble in vivo pseudostratified mucociliary epithelium. Prior to inoculation, the apical surface of well-differentiated HAE cells was washed three times with PBS. The apical surface of cell cultures was inoculated with MASCp36 and its parental isolate WT (BetaCoV/Beijing/IMEBJ05/2020, Nos. GWHACBB01000000) at an MOI of 0.1 at 37 °C for 2 h, and washed three times with PBS to remove the unbounded virus. HAE cells were cultured at an air-liquid interface at 37 °C with 5% CO$_2$. At 24, 48, and 72 h post inoculation, 300 μL of PBS was applied to the apical surface of cell cultures and collected after an incubation for 10 min at 37 °C. Viral RNA copies were quantified by real-time qPCR and shown as mean ± SD. WT or MASCp36-infected cells as well as uninfected cells were fixed with 4% paraformaldehyde, permeabilized with Triton X100, and blocked with 10% BSA. The cells were then incubated with anti-SARS-CoV-2 N protein antibody (Sinobiological, 40143-R004, 1:1000) and HRP-conjugated secondary antibody, followed by DAPI staining. The percentage of SARS-CoV-2 N protein positive cells were shown as mean ± SD ($n = 4$).

**Statistical analysis.** Statistical analyses were carried out using Prism software (GraphPad). All data are presented as means ± standard error of the means. Statistical significance among different groups was calculated using Student's $t$ test, Fisher's exact test, two-way ANOVA, Mann–Whitney test, or Log-rank (Mantel–Cox). *, **, and *** indicate $P < 0.05$, $P < 0.01$, and $P < 0.001$, respectively.

**Reporting summary**. Further information on research design is available in the Nature Research Reporting Summary linked to this article.

## Data availability

The RNA-Seq data generated in this study have been deposited in the NCBI Gene Expression Omnibus (GEO) database under accession code GSE166778. The global cryo-EM maps of RBD$_{MACSp6}$-Fab$_{B8}$-Fab$_{D14}$-hACE2, RBD$_{MACSp25}$-Fab$_{B8}$-Fab$_{D14}$-hACE2, RBD$_{MACSp36}$-Fab$_{B8}$-Fab$_{D14}$-hACE2, and RBD$_{MACSp36}$-Fab$_{B8}$-Fab$_{D14}$-mACE2 complexes are deposited in the Electron Microscopy Data Bank (https://www.ebi.ac.uk/pdbe/emdb/) under accession numbers EMD-31542, EMD-31543, EMD-31544, and EMD-31546, respectively. The determined atomic models of RBD$_{MACSp6}$-Fab$_{B8}$-Fab$_{D14}$-hACE2, RBD$_{MACSp25}$-Fab$_{B8}$-Fab$_{D14}$-hACE2, RBD$_{MACSp36}$-Fab$_{B8}$-Fab$_{D14}$-hACE2, and RBD$_{MACSp36}$-Fab$_{B8}$-Fab$_{D14}$-mACE2 complexes have been deposited to the Protein Data Bank (https://www.rcsb.org/) under accession codes 7FDG, 7FDH, 7FDI, and 7FDK, respectively. All the high throughput sequencing data have been deposited in the Sequence Read Archive (SRA, https://www.ncbi.nlm.nih.gov/sra) of National Center of Biotechnology Information (NCBI), with the BioProject accession number of PRJNA746117. The source data underlying Figs. 1a–h, 2a–c, 3a–c, 4a–d, 5a–e, 6a, b, 7a-d, 8c, d, 9a, b and Supplementary Figs. 1, 2, 3, 4a–c, 5a, b are provided as a Source Data file. Source Data are provided with this paper.

## Materials availability

All requests for resources and reagents should be directed to C.-F.Q. and will be fulfilled after completion of a materials transfer agreement. Source Data are provided with this paper.

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

## Acknowledgements

We thank Dr X. D. Yu and Dr J. J. Zhao for excellent technical and biosafety support. This work was supported by the National Key Plan for Scientific Research and Development of China (2020YFC0841100, 2020YFC0840900, 2020YDC084900, 2020YFA0707500, 2018YFA0900801), the National Science and Technology Major Project of China (No.2017ZX10304402003), the Strategic Priority Research Program (XDB29010000, XDB37030000), the National Natural Science Foundation of China (82041006, 32130005), and the Beijing Municipal Science and Technology Project (No.Z201100001020004). C.-F. Q. was supported by the National Science Fund for Distinguished Young Scholar (No. 81925025), and the Innovative Research Group (No. 81621005) from the NSFC, and the Innovation Fund for Medical Sciences (No.2019RU040) from the Chinese Academy of Medical Sciences (CAMS). Xiangxi Wang was supported by Ten Thousand Talent Program and the NSFS Innovative Research Group (No. 81921005).

## Author contributions

S.S., H.G., L.C., Q.C., Q.Y., G.Y., R.-T.L., H.F., Y.-Q.D., X.S., Y.Q., M.L., J.L., R.F., Y.G., N.Z., S.Q., L.W., Y.-F.Z., C.Z., L.Z., Yue. C., Yu. C., X.W., and M.S. performed experiments; S.S., H.G., Q.C., G.Y., R.-T. L., H.F, Yu. C., X.Y., X.W., W.T., and H.W. analyzed data. C.-F.Q., X.W., H.G., and S.S conceived the project and designed the experiments. S.S. and H.G., and R.-T.L. wrote the draft of the manuscript. C.-F.Q., and X.W. supervised the study and wrote the manuscript with the input of all co-authors.

## Competing interests

The authors declare no competing interests.
