## [Peer Review File · Nature Communications]

REVIEWER COMMENTS

Reviewer #1 (Remarks to the Author):

A tractable mouse model is an important tool to evaluate antiviral therapies and understanding of pathogenesis of SARS-CoV-2. Numerous mouse models have been developed for this purpose such as AdV, AAV, transgenetically or knockin human ACE2 into mouse in vivo. However, these models are not convenient for pathogenesis or immunological study, as it will take long time to cross the humanized ACE2 mouse and specific gene knockout mouse. In this study, Cheng-Feng and colleagues obtained a mouse-adapted SARS-CoV-2 (MASCp36) based on MASCp6. Different from MASCp6, the MASCp36 carried more adaptive mutations and the p36 virus infection leads to a lethal phenotype. The authors also performed detailed mechanistic studies to reveal the adaptive mechanism that the mutations in the RBD domain could render the capability of the adapted virus to utilize mouse ACE2 as the receptor. This manuscript is well-written, concise, and the experimentations are well designed. I have only have one question and other minor concerns, typos to be raised.

Major question:

1. The adapted virus adapt the mouse ACE2 as the receptor, and it is important to check whether the adapted virus can still use human ACE2 as the receptor. This is very important for this model to evaluate the monoclonal antibody since there are mutations in the RBD domain of mouse adapted virus.

Minor concerns, typos

1. The citation of the paper should be carefully examined. I found few mistakes, but maybe not limited to these. For examples: "AAV-hACE2 transduced mice¹¹ and Ad5-hACE2 transduced mice¹² had been developed. Furthermore, mouse adapted strains of SARS-CoV-2 have also been developed via either in vivo passaging or reverse genetics^{13,14}." The AAV-Hace2 model was developed by Iwasaki lab published in JEM, the author did not cite the reference correctly. Reference #12 is not the Ad5-hACE2 model, in addition, reference 14 is not the study that utilization of reverse genetics to generate mouse adapted stain of SARS-CoV-2. Ralf Baric et al used reverse genetics to generate the mouse adapted virus, which is in Nature. Anyway, I will suggest the authors to double check all the references to avoid the misleading.
2. "In our previous study, we generated a mouse adapted strain of SARS-CoV-2 (MASCp6) by 6 serial passages of a SARS-CoV-2 in the lung of aged BALB/c mice, which cause moderate lung damage and no fatality in mice." The authors should cite their previous studies here.
3. "We further characterized the in vivo replication dynamics of MASCp6 in both young and aged mice," Is it a typo here? Maybe the authors meant p36 here, please check it.
4. Maybe not related to this study, but I suggest the authors to add page number and line number in the future, which will be more convenient for the reviewers to make comments by citation of the page number and line number.

Reviewer #2 (Remarks to the Author):

'Characterization and structural basis of a lethal mouse-adapted SARS-CoV-2' by Shihui Sun et al describes development of a mouse adapted, lethal SARS-CoV-2 that causes severe lung disease in aged mice. This work builds on a previous report of a minimally mouse adapted virus and presents a useful tool to the virology community. In addition to the disease characterization, the structural and binding data is a strength of this work and helps our understanding of ACE2-spike interactions. The authors identify N501Y, Q493H, and K417N as critical mutations that subsequently emerged at the receptor binding domain (RBD) of MASCp36 and improve binding. While the data is compelling, more information is needed in several areas to fully define this model and justify some of the claims that are made. In particular, data from later timepoints and virus replication data are essential to understand the MASC36 model. The authors should also be careful about their definition of old mice and give more information about the non-spike mutations that arose during mouse adaptation.

General comments:

Introduction, text after figure 4 - It is incorrect to state that no COVID-19 mouse model results in severe lung disease. Leist et al demonstrated severe ALI and lethal respiratory disease.

Authors should be careful about age descriptions. 9 month old mice are middle aged, not old. Truly aged mice are >18 months old per the National Institute of Aging.

Major points:

It is inappropriate to share video of distressed laboratory animals, this is not suitable for public release.

Figure 1 - Need replicating virus, either PFU or TCID50 and not just RNA levels.

What about titer or histology at timepoints after 4 dpi?

Extended figure 6C – The authors should show the 45 cytokines/chemokines as log₂FC as well as the less common TPM presentation.

6D – Luminex data should be quantified, not given in a heat map without scale.

Cytokine/chemokine data in general – is any data available from an earlier timepoint? D4 is late to be examining the innate immune response and some of the mice would have already been dying by this timepoint.

Figure 3 – when were the mice given antibody relative to the time of infection? The methods list two times but there is only 1 antibody group in the figure.

Figure 3b – Please clarify the timepoint in the text and figure.

With no histology data past D4 it is unsurprising that this model does not show any signs of pulmonary fibrosis. However since COVID-19 patients do show signs of fibrosis, the authors should look at mice that survive at D7 and beyond for signs of tissue regeneration and/or fibrosis.

Page 10 – In the first few sentences the authors discuss immunological contrasts between young and old mice in the MASC26 model without showing or citing any data. Please show the data regarding D1 immune cell infiltrates or remove the text.

Minor points:

Figure 1a-d – it would be easier to understand this figure if the dose colors on the graphs were consistent.

Final full paragraph on page 9 – the final sentence is unclear and needs rephrasing.

The authors refer to 'sticky secretions' in a couple of instances without any definition, what does this mean?

In discussions of coronavirus sex differences it would be appropriate to consider and discuss the work of Channappanavar, et al 2018, PMID 28373583 that examined sex differences in a SARS-CoV mouse model

The authors should devote more time/space discussing the MASC36 mutations that developed outside of the spike gene and how they might contribute to virulence.

Was the final MASC36 virus plaque purified before sequencing and further use?

When did the non-spike mutations arise during passage?

Line numbers would make review much easier.

reference 12 in the introduction is incorrect, this paper does not use Ad5-hACE2 transduced mice

Reviewer #3 (Remarks to the Author):

In this manuscript the authors describe further mouse passaging of their initial mouse-adapted strain of SARS-CoV-2. This new strain shows age and gender-dependent mortality similar to human infection, viral replication limited to the trachea and lung, and more serious lung damage than reported in previous studies. This is an important advancement as it better models human

disease.

Major points

- Fig 1e and f are labeled sgRNA but use the same primers as the original study, which is labeled gRNA – so something is mislabeled. Also there is no explanation of how viral load is being calculated. In addition it would be important to show actual viral titers at least in the lung of p6, p25, and p36 infected mice to show that you are getting a real increase in viral replication with serial passage.
- The authors claim the mice develop ARDS but it is unclear what the evidence is to support this claim. They show lung damage by histology but would need to show measures of lung function to support the ARDS claim.
- The authors show an important difference between male and female, and aged and young mice but provide no explanation for the differences. It would be nice to see ACE2 levels measured by qPCR or western blot in these different groups. In addition, given the large differences between male and female mice, all figure legends should include the sex of the animals. They discuss sex differences in innate immunity but do not provide any data supporting this in their model.
- In figure 1g the ACE2 expressing cells do not appear to be infected. What dose of virus was used and what is the explanation for lack of infection of ACE2+ cells?
- Extended Fig 2 refers to framed areas that I don't see in the top image. It's also not clear what area you are quantifying in this figure.
- Given the recent emergence of N501Y in the human population, some explanation should be provided as to how this mutation helps with both binding to human and mouse ACE2.

Minor points:

- p4 – include reference to previous study
- p5 – MASCP6 – should be p36?
- "with bilateral cardinal red appearance" – should be bilateral
- Consequently, a striking loss of SPC+ AT2 cells with apoptosis were observed in the lung from aged mice. This doesn't make sense – reword sentence.
- p9 – "cross-transmission of SARS-CoV-2" don't know what this means and there is no transmission data in the paper.
- "Multiplex immunofluorescence staining confirmed H014 treatment completely protected animals from viral infection and replication by preventing AT2 loss and neutrophil infiltration caused as a result of MASCP36 infection" – the wording of this sentence doesn't make sense, the H014 treatment prevented viral infection/replication, which subsequently protected animals from AT2 loss and neutrophil infiltration.
- Extended data 6 and 7 are quite important in terms of showing that this virus models human disease well in terms of cytokine/chemokine induction and macrophage/neutrophil infiltration – I would think this should be in the main text figures.

Reviewer #4 (Remarks to the Author):

COVID-19 caused by SARS-CoV-2 is a severe threat to public health worldwide. An efficient small animal model is an urgent need for the study on SARS-CoV-2. As the wild type SARS-CoV-2 does not infect mouse, several transgenic mouse models were developed in previous studies, which can only exhibit mild to moderate lung damage upon challenge by SARS-CoV-2. Other than the transgenic mouse model, the authors developed a mouse-adapted SARS-CoV-2 strain MASCP36 using in vivo passaging methods, which causes the symptoms of COVID-19 in animal model, especially reproducing the age- and sex-dependent mortality of human COVID-19. Deep-sequencing identified mutations in the genome of MASCP36, including three amino acid substitutions in RBD of MASCP36 that are crucial for the binding to mACE2. The authors solved the cryo-EM structure of mACE2 in complex with the RBD of MASCP36, revealing the molecular basis for the host transition driven by the amino acid substitutions. The mouse-adapted SARS-CoV-2 strain MASCP36 presented in this work has a great value for studying the pathogenesis of COVID-19 and evaluating vaccines and therapeutics against COVID-19. The followings are the points for this manuscript that should be addressed.

Major points:

1. From the perspective of experimental and biological safety, can MASCP36 bind ACE2 and/or infect cells of human and other animals? The binding between RBD of MASCP36 and hACE2 should be tested and measured.
2. The mutations of virus can cause immune escape. Do these mutations affect the evaluation of vaccines and therapeutics using MASCP36?
3. The authors need to discuss why H014 preserves its protective ability against virus.
4. More therapeutics against COVID-19 other than H014 need to be tested.
5. What is the standard to define a mouse as aged? The authors used 9-month mice in the experiments. How old are these mice as old as humans? The authors need to specify this in the main text.
6. Sequence alignment of ACE2 from different species is required to show the adaptation of MASCP36 to mACE2 from hACE2.

Minor points:

7. How is PFU defined?
8. what is SARS-CoV MA15 ?

Reviewer #1 (Remarks to the Author):

A tractable mouse model is an important tool to evaluate antiviral therapies and understanding of pathogenesis of SARS-CoV-2. Numerous mouse models have been developed for this purpose such as AdV, AAV, transgenetically or knockin human ACE2 into mouse in vivo. However, these models are not convenient for pathogenesis or immunological study, as it will take long time to cross the humanized ACE2 mouse and specific gene knockout mouse. In this study, Cheng-Feng and colleagues obtained a mouse-adapted SARS-CoV-2 (MASCp36) based on MASCp6. Different from MASCp6, the MASCp36 carried more adaptive mutations and the p36 virus infection leads to a lethal phenotype. The authors also performed detailed mechanistic studies to reveal the adaptive mechanism that the mutations in the RBD domain could render the capability of the adapted virus to utilize mouse ACE2 as the receptor. This manuscript is well-written, concise, and the experimentations are well designed. I have only have one question and other minor concerns, typos to be raised.

Major question:

1. The adapted virus adapt the mouse ACE2 as the receptor, and it is important to check whether the adapted virus can still use human ACE2 as the receptor. This is very important for this model to evaluate the monoclonal antibody since there are mutations in the RBD domain of mouse adapted virus.

Response: Thank for the reviewer's insightful comments. We have now included two panel of experiments to make sure whether the mouse adapted virus MASCp36 can still use human ACE2 as the receptor: 1) Binding affinity assay with hACE2; 2) In vitro infectivity assay with HAE culture. All these new results showed MASCp36 retained the capability to utilize hACE2, and these new results were shown **in new Fig. 10a-c**.

Minor concerns, typos

1. The citation of the paper should be carefully examined. I found few mistakes, but maybe not limited to these. For examples: "AAV-hACE2 transduced mice¹¹ and Ad5-hACE2 transduced mice¹² had been developed. Furthermore, mouse adapted strains of SARS-CoV-2 have also been developed via either in vivo passaging or reverse genetics^{13,14}." The AAV-Hace2 model was developed by Iwasaki lab published in JEM, the author did not cite the reference correctly. Reference #12 is not the Ad5-hACE2 model, in addition, reference 14 is not the study that utilization of reverse genetics to generate mouse adapted stain of SARS-CoV-2. Ralf Baric et al used reverse genetics to generate the mouse adapted virus, which is in Nature. Anyway, I will suggest the authors to double check all the references to avoid the misleading.

Response: All corrected.

2. "In our previous study, we generated a mouse adapted strain of SARS-CoV-2 (MASCp6) by 6 serial passages of a SARS-CoV-2 in the lung of aged BALB/c mice, which cause moderate lung damage and no fatality in mice." The authors should cite their previous studies here.

Response: Corrected.

3. "We further characterized the in vivo replication dynamics of MASCp6 in both young and aged mice," Is it a typo here? Maybe the authors meant p36 here, please check it.

Response: Corrected.

4. Maybe not related to this study, but I suggest the authors to add page number and line number in the future, which

will be more convenient for the reviewers to make comments by citation of the page number and line number.

Response: Thanks. The page number and line numbers have been updated in our revised manuscript.

Reviewer #2 (Remarks to the Author):

'Characterization and structural basis of a lethal mouse-adapted SARS-CoV-2' by Shihui Sun et al describes development of a mouse adapted, lethal SARS-CoV-2 that causes severe lung disease in aged mice. This work builds on a previous report of a minimally mouse adapted virus and presents a useful tool to the virology community. In addition to the disease characterization, the structural and binding data is a strength of this work and helps our understanding of ACE2-spike interactions. The authors identify N501Y, Q493H, and K417N as critical mutations that subsequently emerged at the receptor binding domain (RBD) of MASCP36 and improve binding. While the data is compelling, more information is needed in several areas to fully define this model and justify some of the claims that are made. In particular, data from later timepoints and virus replication data are essential to understand the MASCP36 model. The authors should also be careful about their definition of old mice and give more information about the non-spike mutations that arose during mouse adaptation.

General comments:

Introduction, text after figure 4 - It is incorrect to state that no COVID-19 mouse model results in severe lung disease. Leist et al demonstrated severe ALI and lethal respiratory disease.

Response: Thanks for the comments. We have updated this information and cited Leist et al's paper accordingly.

Authors should be careful about age descriptions. 9 month old mice are middle aged, not old. Truly aged mice are >18 months old per the National Institute of Aging.

Response: Thanks for the reviewer's suggestion. We totally agree with the reviewer's point. To be accurate, we have changed the description of "young" and "aged" with the exact description "8-week-old" and "9-month-old" in the revised manuscript.

Major points:

It is inappropriate to share video of distressed laboratory animals, this is not suitable for public release.

Response: Thanks for the suggestion. We have deleted the video accordingly.

Figure 1 - Need replicating virus, either PFU or TCID50 and not just RNA levels.

Response: Thanks for the kind suggestion. We have now performed standard plaque forming assay with lung homogenate from the MASCP36-infected animals, and the new results were included as **new Fig. 1 g-h**.

What about titer or histology at time points after 4 dpi?

Response: Thanks for the question. In our study, all 9-month-old male mice succumbed to MASCP36 infection (12,000 PFU) within 4 days, so no viral titer or pathology results was obtained at time points after 4 dpi. Herein, to observe the potential impact at late stage of infection, we further performed additional challenge experiment with a lower dose (120 PFU), and the viral replication and lung histology was performed as suggested. Under this condition, high level viral sgRNA ($10^{7.04 \pm 0.28}$ copies /g) were detected in lung tissues, and lung fibrosis was observed in the MASCP36-infected mice as evidenced by the depositions of collagen in pulmonary artery wall and thickened alveoli. These new data was combined as new **Fig 3c and Supplementary Fig 2**.

Extended figure 6C – The authors should show the 45 cytokines/chemokines as log2FC as well as the less common TPM presentation.

Response: Thanks for your comment. We have replaced the corresponding data into log2FC style, and the data collected at 1dpi from both 9-month-old and 8-week-old male mice were also included in current version as new **Fig 5c**.

6D – Luminex data should be quantified, not given in a heat map without scale.

Response: Thanks, we showed the quantified Luminex data in new **Fig. 5d**.

Cytokine/chemokine data in general – is any data available from an earlier timepoint? D4 is late to be examining the innate immune response and some of the mice would have already been dying by this timepoint.

Response: Thanks for the insightful comments. We totally agree with the reviewer that an earlier time point would provide more information. Accordingly, we have performed RNA-seq with lung samples at 1 dpi and 4 dpi. Our analysis showed that MASCP36 infection induced robust innate immune response at 1dpi in 9-month-old mice, and T cell activation in 9-month-old mice was not as strong as that in young mice at 4 dpi. Meanwhile, we found genes involved in “cilium movement” were significantly stimulated at 1 and 4 dpi in 8-week-old mice, while much less genes related to "cilium movement" were upregulated in 9-month-old mice till 4 dpi. These results help to understand the age-skewed mortality observed in our model, and all these results were combined in the new **Fig. 5**.

We also discuss the potential implication of these results in the Discussion. Previous studies show that mucociliary clearance represents an important defense mechanism in the respiratory tract that requires coordinated ciliary activity and proper mucus production to propel airway surface liquids that traps pathogens and pollutants, permitting their clearance from the lungs (Kamiya et al., 2020). A recent study shows that ACE2 receptor protein robustly localizes within the motile cilia of airway epithelial cells, which likely represents the initial or early subcellular site of SARS-CoV-2 viral entry during host respiratory transmission (Lee et al., 2020). So, we supposed that the stimulation of cilium movement in younger mice contribute to a rapid mucociliary clearance of virus at the early stage of MASCP36 infection, while over-stimulated innate immune response and reduced adaptive immune response in the elder mice explained the higher mortality in response to MASCP36 challenge. We have also added this discussion in the revised manuscript.

Figure 3 – when were the mice given antibody relative to the time of infection? The methods list two times but there is only 1 antibody group in the Figure 3b – Please clarify the timepoint in the text and figure.

Response: The mice were given antibody treatment two times at 24 hours before and after MASCP36 infection. The administration procedure is the same as our previous paper (Lv, et al. Science, 2020) using another mouse model.

With no histology data past D4 it is unsurprising that this model does not show any signs of pulmonary fibrosis. However since COVID-19 patients do show signs of fibrosis, the authors should look at mice that survive at D7 and beyond for signs of tissue regeneration and/or fibrosis.

Response: As responded above, we have now included new data collected at 7dpi . The results showed that the lung tissue had signs of fibrosis by Masson’s trichrome staining (new **Fig.3c**), and of alveolar mesenchymal cell proliferation detected by multiplex immunofluorescence staining of Ki67 and Vimentin (new **Supplementary Fig 2**). The results demonstrated that the mice model presented many clinical signs including pulmonary fibrosis.

Page 10 – In the first few sentences the authors discuss immunological contrasts between young and old mice in the MASCP36 model without showing or citing any data. Please show the data regarding D1 immune cell infiltrates or remove the text.

Response: Thanks for the comments. We have now included new data comparing the infiltration of macrophages and neutrophils on day 1 and 4 upon MASCp36 infection as new **Fig. 6b**.

Minor points:

Figure 1a-d – it would be easier to understand this figure if the dose colors on the graphs were consistent.

Response: Thanks for the kind suggestions. We have now unified the dose colors in **Fig 1 a-d**.

Final full paragraph on page 9 – the final sentence is unclear and needs rephrasing.

Response: Corrected.

The authors refer to 'sticky secretions' in a couple of instances without any definition, what does this mean?

Response: Sorry for the unclear description. We have changed into "sticky mucus" in revision.

In discussions of coronavirus sex differences it would be appropriate to consider and discuss the work of Channappanavar, et al 2018, PMID 28373583 that examined sex differences in a SARS-CoV mouse model.

Response: Thanks for the reviewer's suggestion. We cited and discussed the corresponding section (Channappanavar, et al 2018, PMID 28373583) as the following: Channappanavar, et al. had studied that estrogen receptor signaling is critical for protection in females against SARS-CoV infection. So, we speculated here that estrogen may also play an important role in the protection against MASCp36 infection in female mouse model. We added it in our revised manuscript.

The authors should devote more time/space discussing the MASC36 mutations that developed outside of the spike gene and how they might contribute to virulence.

Response: Thanks for your suggestion. We totally agree with the reviewer that other mutations outside of the spike gene might also play role. We have added discussion about outside of the S gene and their contribution to virulence.

Was the final MASC36 virus plaque purified before sequencing and further use?

Response: The working stock of MASCp36 was titerated by plaque assay and sequenced.

When did the non-spike mutations arise during passage?

Response: There are 9 mutations outside of S. The exact passages when these mutations emerged were summarized in the following table and shown in **Fig. 8a**.

Mutation site	Located gene	Time
D128Y	N gene	MASCp6
R32C		MASCp36
L37F	Nsp6	MASCp6
A128V		MASCp15
P84S	Nsp10	MASCp6
I1258V	Nsp3	MASCp36
H470Y	Nsp4	MASCp30
S8F	Nsp8	MASCp30
S301L	Nsp5	MASCp36

Line numbers would make review much easier. Reference 12 in the introduction is incorrect, this paper does not use Ad5-hACE2 transduced mice

Response: Line numbers were added and reference 12 were correct in the revised manuscript.

Reviewer #3 (Remarks to the Author):

In this manuscript the authors describe further mouse passaging of their initial mouse-adapted strain of SARS-CoV-2. This new strain shows age and gender-dependent mortality similar to human infection, viral replication limited to the trachea and lung, and more serious lung damage than reported in previous studies. This is an important advancement as it better models human disease.

Major points

• *Fig 1e and f are labeled sgRNA but use the same primers as the original study, which is labeled gRNA – so something is mislabeled. Also there is no explanation of how viral load is being calculated. In addition it would be important to show actual viral titers at least in the lung of p6, p25, and p36 infected mice to show that you are getting a real increase in viral replication with serial passage.*

Response: We used sgRNA primers the same as previously described in Roman Wölfel et al, 2020 May;581(7809):465-469. The viral load were calculated based on standard curve, and we have mentioned this in the material and methods section.

Thanks for the suggestion, we compared viral titers in the lung of p6 and p36 infected mice. Indeed, there is an increase in viral titer, which was in agreement with the SPR assay and structure analysis.

• *The authors claim the mice develop ARDS but it is unclear what the evidence is to support this claim. They show lung damage by histology but would need to show measures of lung function to support the ARDS claim.*

Response: Thanks for the reviewer's suggestion. The Acute Respiratory Distress Syndrome (ARDS) is a common clinical syndrome of acute lung inflammation, non-cardiogenic pulmonary edema and acute respiratory failure ¹. According to the definition of ARDS by the American European Consensus Conference (AECC) in 1994 ² or the Berlin Definition in 2012 ³, not only lung damage by histology, but also other clinical hallmarks about lung function, such as hypoxemia or decreased lung compliance, are critical for the evaluation of ARDS. Nevertheless, because of the limitation of detecting instrument in our P3 level laboratory, the lung function can't be detected here although we observed dyspnea in MASCp36 infected mice. So, to describe accurately and prudently the clinical syndrome of this mouse model, we deleted the claim of ARDS in our revised manuscript.

References:

1. Ware LB, Matthay MA: Medical progress: the acute respiratory distress syndrome. N Engl J Med 2000, 342:1334–1349.
2. Bernard GR, Artigas A, Brigham KL, Carlet J, Falke K, Hudson L, Lamy M, Legall JR, Morris A, Spragg R: The American-European Consensus Conference on ARDS. Definitions, mechanisms, relevant outcomes, and clinical trial coordination. Am J Respir Crit Care Med 1994, 149:818–824.
3. Ranieri VM, Rubenfeld GD, Thompson BT, Ferguson ND, Caldwell E, Fan E, Camporota L, Slutsky AS: Acute respiratory distress syndrome: the Berlin Definition. JAMA 2012, 307:2526–2533.

• *The authors show an important difference between male and female, and aged and young mice but provide no explanation for the differences. It would be nice to see ACE2 levels measured by qPCR or western blot in these different groups. In addition, given the large differences between male and female mice, all figure legends should include the sex of the animals. They discuss sex differences in innate immunity but do not provide any data supporting this in their model.*

Response: Thanks for your suggestion. To understand the gender- and sex-skewed mortality in mice, we performed RNA-seq analysis with lung homogenates from mice at different age and gender. These new results, including ACE2 comparison, have been combined into revision in **new Fig.5a-e and supplementary Fig. 4a**. Some of the results were further validated by Luminex analysis (**supplementary Fig.3**) and immunostaining assay (**Fig.6a and b**). Additionally, we have indicated the sex of mice in all figure legends.

• *In figure 1g the ACE2 expressing cells do not appear to be infected. What dose of virus was used and what is the explanation for lack of infection of ACE2+ cells?*

Response: In original figure 1g (**now Fig.2a**), the ACE2 expression was shown by the immunofluorescence staining in lung tissues of SARS-CoV-2 infected mice (12,000PFU/mouse). The result showed that ACE2 expression decreased upon infection, which was similar to mouse infected with SARS-CoV (Josef M Penninger, Nat Med, 2005). Importantly, SARS-CoV or SARS-CoV-2 infection could downregulate ACE2 expression and/or induce apoptosis, thus further decreased the number of ACE2+SARS-CoV-2+ cells. We have added colored arrows to clearly indicate the infected cells with or without ACE2 expression, and moved these data to Fig. 2a.

• *Extended Fig 2 refers to framed areas that I don't see in the top image. It's also not clear what area you are quantifying in this figure.*

Response: Thanks for the reviewer's suggestion, and we added the frame in the **revised Fig 2c**. We quantified all the ACE2+SPC+-double positive cells in the SPC+ AT2 compartment within the whole lung sections.

• *Given the recent emergence of N501Y in the human population, some explanation should be provided as to how this mutation helps with both binding to human and mouse ACE2.*

Response: Previous studies showed that N501 forms part of the binding loop in the contact region of hACE2, forming a hydrogen bond with Y41 in hACE2¹, which stabilized via K353 binding hotspot residues on hACE2³⁴. It contributes to the enhanced binding affinity of SARS-CoV-2 for hACE2³.

In addition, we found that N501Y increased the binding affinity of mouse ACE2 by structural remodeling.

References:

1. Lan J, Ge J, Yu J, et al. Structure of the SARS-CoV-2 spike receptor-binding domain bound to the ACE2 receptor. Nature. 2020;581(7807):215-220.
2. Wang Y, Liu M, Gao J. Enhanced receptor binding of SARS-CoV-2 through networks of hydrogen-bonding and hydrophobic interactions. Proc Natl Acad Sci U S A. 2020;117(25):13967-13974.
3. Shang J, Ye G, Shi K, et al. Structural basis of receptor recognition by SARS-CoV-2. Nature. 2020;581(7807):221-224.

Minor points:

• *p4 – include reference to previous study*

Response: We cited our previous studies here in the revised manuscript.

• *p5 – MASCP6 – should be p36?*

Response: Yes, we meant MASCP36 and revised it in the revised manuscript.

- *“with biolateral cardinal red appearance” – should be bilateral*

Response: Corrected.

- *Consequently, a striking loss of SPC+ AT2 cells with apoptosis were observed in the lung from aged mice. This doesn't make sense – reword sentence.*

Response: We reworded the sentence to “Consequently, a striking apoptosis of AT2 cells were observed in the lung from 9-month-old mice”

- *p9 – “cross*

-transmission of SARS-CoV-2” don't know what this means and there is no transmission data in the paper.

Response: “cross-transmission” was deleted in the revised manuscript.

- *“Multiplex immunofluorescence staining confirmed H014 treatment completely protected animals from viral infection and replication by preventing AT2 loss and neutrophil infiltration caused as a result of MASCP36 infection” – the wording of this sentence doesn't make sense, the H014 treatment prevented viral infection/replication, which subsequently protected animals from AT2 loss and neutrophil infiltration.*

Response: We revised the sentence according to the reviewer's suggestion.

- *Extended data 6 and 7 are quite important in terms of showing that this virus models human disease well in terms of cytokine/chemokine induction and macrophage/neutrophil infiltration – I would think this should be in the main text figures.*

Response: Thanks for your suggestion. We revised **Extended data 6 and 7 as Fig5 and 6** in the main text figures.

Reviewer #4 (Remarks to the Author):

COVID-19 caused by SARS-CoV-2 is a severe threat to public health worldwide. An efficient small animal model is an urgent need for the study on SARS-CoV-2. As the wild type SARS-CoV-2 does not infect mouse, several transgenic mouse models were developed in previous studies, which can only exhibit mild to moderate lung damage upon challenge by SARS-CoV-2. Other than the transgenic mouse model, the authors developed a mouse-adapted SARS-CoV-2 strain MASCP36 using in vivo passaging methods, which causes the symptoms of COVID-19 in animal model, especially reproducing the age- and sex-dependent mortality of human COVID-19. Deep-sequencing identified mutations in the genome of MASCP36, including three amino acid substitutions in RBD of MASCP36 that are crucial for the binding to mACE2. The authors solved the cryo-EM structure of mACE2 in complex with the RBD of MASCP36, revealing the molecular basis for the host transition driven by the amino acid substitutions. The mouse-adapted SARS-CoV-2 strain MASCP36 presented in this work has a great value for studying the pathogenesis of COVID-19 and evaluating vaccines and therapeutics against COVID-19. The followings are the points for this manuscript that should be addressed.

Response: Many thanks for the very positive comments.

Major points:

1. From the perspective of experimental and biological safety, can MASCP36 bind ACE2 and/or infect cells of human and other animals? The binding between RBD of MASCP36 and hACE2 should be tested and measured.

Response: Thanks for your constructive suggestion. As responded above, we have now included two panels of experiments: 1) We measured the binding affinity between RBD of MASCP36 and hACE2; 2) We assayed the in vitro infectivity with HAE. All these new results suggested that MASCP36 didn't show enhanced infectivity to human and the new data are combined in new **Fig. 9 and 10** in revision.

2. The mutations of virus can cause immune escape. Do these mutations affect the evaluation of vaccines and therapeutics using MASCP36?

Response: Thanks for your comments. We agree with the reviewer that some specific mutation in S protein might cause immune escapee, and our MASCP36 model may not be applied to all antibody or vaccine evaluation. In our study, we noted H014 conferred complete protection, which was in agreement with previous results and epitope mapping (Lv, et al, Science, 2020). As MASCP36 contained specific mutation in the RBD, mAbs that targets these epitopes may not be applicable for this model. Interestingly, these specific mutations, including N501Y and K417N, have been recorded in human variants and became predominant. We have mentioned this in the revised manuscript.

3. The authors need to discuss why H014 preserves its protective ability against virus.

Response: Thanks for the comments. As we have determined the epitopes of H014 in our previous publication (Lv, et al, Science, 2020), and all epitopes are not relevant to the mutation identified in our adapted strain MASCP36. That's why H014 retained the protection efficacy.

4. More therapeutics against COVID-19 other than H014 need to be tested.

Response: Thanks for the suggestion. We are happy to include more validation data using this model, however, current version manuscript contained 10 main figures and 12 supplementary items, which has approached the maximal requirement of Nature Communications. Considering our present study is to establish and characterize the

model, we are wondering the validation of other therapeutics should be published in separate articles. Several manuscripts using our model are being prepared.

5. *What is the standard to define a mouse as aged? The authors used 9-month mice in the experiments. How old are these mice as old as humans? The authors need to specify this in the main text.*

Response: Thanks for the comments. 9-month-old mice, usually considered as the middle-aged mice, were used based on our previous experience from SARS and MERS-CoV adaption studies. To be accurate, we have replaced "aged" into "9-month-old" throughout the manuscript.

6. *Sequence alignment of ACE2 from different species is required to show the adaptation of MASCP36 to mACE2 from hACE2.*

Response: Thanks for the comments. As several papers have extensively compared the sequence alignment of ACE2 from different species (Min Li et.al, *Brief Bioinform*, 2021; Chao Qin et.al, *Cell Discovey*; Qiang Ding et.al, *PNAS*, 2021) . we didn't show this result again. Herein, we have now included Cryo-EM data and atomic model refinement statistics of RBD mutant-mACE2 and RBD mutant-hACE2 complex as new **Supplementary Table 1 and 3** to explain the adaptation process.

Minor points:

7. *How is PFU defined?*

Response: PFU means "plaque forming unit".

8. *what is SARS-CoV MA15?*

Response: MA15 is a well-known mouse-adapted strain of SARS-CoV, which was originally developed by serial passaging of wild type SARS-CoV in aged mice for 15 passages (Roberts et al. *PLoS Pathogen*, 2007), we have now cited the original paper and updated the reference list accordingly.

REVIEWERS' COMMENTS

Reviewer #1 (Remarks to the Author):

The authors address all the questions and concerns. The revised manuscript is suitable for publication in Nature Communications.

Reviewer #4 (Remarks to the Author):

The authors have well addressed the issues raised in previous review round. The current manuscript is suitable for acceptance.